# CoSDA: Continual Source-Free Domain Adaptation

## Abstract

Without access to the source data, source-free domain adaptation (SFDA) transfers knowledge from a source-domain trained model to target domains. Recently, SFDA has gained popularity due to the need to protect the data privacy of the source domain, but it suffers from catastrophic forgetting on the source domain due to the lack of data. To systematically investigate the mechanism of catastrophic forgetting, we first reimplement previous SFDA approaches within a unified framework and evaluate them on four benchmarks. We observe that there is a trade-off between adaptation gain and forgetting loss, which motivates us to design a consistency regularization to mitigate forgetting. In particular, we propose a continual source-free domain adaptation approach named CoSDA, which employs a dual-speed optimized teacher-student model pair and is equipped with consistency learning capability. Our experiments demonstrate that CoSDA outperforms state-of-the-art approaches in continuous adaptation. Notably, our CoSDA can also be integrated with other SFDA methods to alleviate forgetting.

## 1 Introduction

Domain adaptation (DA) (Ben-David et al., 2010) aims to transfer features from a fully-labeled source domain to multiple unlabeled target domains. Prevailing DA methods perform the knowledge transfer by consolidating data from various domains and minimizing the domain distance (Ganin et al., 2016; Hoffman et al., 2018; Long et al., 2015; Saito et al., 2018). However, due to the privacy policy, we cannot access source domain data in most cases, where all data and computations must remain local and only the trained model is available (Al-Rubaie & Chang, 2019; Mohassel & Zhang, 2017).

Source-free domain adaptation (SFDA) (Kundu et al., 2020; Li et al., 2020; Liang et al., 2020; 2022b) maintains the confidentiality of the domain data by transferring knowledge straight from a source-domain-trained model to target domains. SFDA also allows for spatio-temporal separation of the adaptation process since the model-training on source domain is independent of the knowledge transfer on target domain. However, due to the lack of alignment with prior domain features, typical SFDA methods tend to overfit the current domain, resulting in catastrophic forgetting on the previous domains (Bobu et al., 2018; Tang et al., 2021; Yang et al., 2021a). This forgetting can lead to severe reliability and security issues in many practical scenarios such as autonomous driving (Shaheen et al., 2022) and robotics applications (Lesort et al., 2020). To address this issue, a possible solution is to preserve a distinct model for each domain, but this solution is impractical since (1) the model pool expands with the addition of new domains, and (2) obtaining the specific domain ID for each test sample is hard.

In this paper, we introduce a practical DA task named continual source-free domain adaptation (continual SFDA), with the primary goal of maintaining the model performance on all domains encountered during adaptation. The settings of continual SFDA are presented in Figure 1I. We initiate the adaptation process by training a model in the fully-labeled source domain, and then subsequently transfer this off-the-shelf model in a sequential manner to each of the target domains. During the testing phase, data is randomly sampled from previously encountered domains, thereby rendering it impossible to determine the specific domain ID in advance.

To systematically investigate the mechanism of catastrophic forgetting, we reimplement previous SFDA approaches within a unified framework and conduct a realistic evaluation of these methods under the

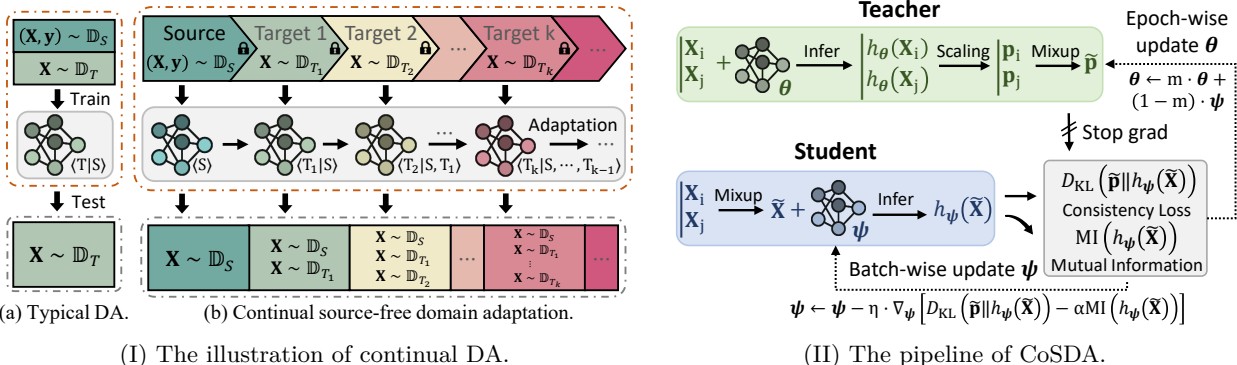

(a) Typical DA.      (b) Continual source-free domain adaptation.

(I) The illustration of continual DA.      (II) The pipeline of CoSDA.

Figure 1: Illustration of continuous source-free domain adaptation. Left: Comparing typical DA (a) and continuous DA (b). In typical DA, models are trained on both source and target domains, but tested only on the target domain. In contrast, continuous DA sequentially trains on each target domain and tests on all previously seen domains. Right: The pipeline of the proposed CoSDA method, utilizing a dual-speed optimized teacher-student model pair to adapt to new domains while avoiding forgetting.

continual SFDA settings on four multi-domain adaptation benchmarks, i.e. DomainNet (Peng et al., 2019), Office31 (Saenko et al., 2010), OfficeHome (Venkateswara et al., 2017) and VisDA (Peng et al., 2017). To ensure the representativeness of our evaluation, we select six commonly used SFDA methods as follows: SHOT (Liang et al., 2020), SHOT++ (Liang et al., 2022b), NRC (Yang et al., 2021b), AaD (Yang et al., 2022), DaC (Zhang et al., 2022) and EdgeMix (Kundu et al., 2022). For further comparison, we also consider two well-performed continual DA methods: GSFDA (Yang et al., 2021a) and CoTTA (Wang et al., 2022). We measure the extent of forgetting exhibited by the aforementioned methods in both single-target and multi-target sequential adaptation scenarios.

As shown in Figure 2, our experiments reveal two main findings: (1) the accuracy gain in the target domain often comes at the cost of huge forgetting in the source domain, especially for hard domains like quickdraw; (2) the catastrophic forgetting can be alleviated with data augmentations (e.g., DaC and Edgemix) and domain information preservation (e.g., GSFDA and CoTTA). Our investigation also finds some limitations of current continual DA techniques, such as GSFDA, which relies on domain ID information for each sample during testing, and CoTTA, which has a tendency to overfit the source domain and learn less plastic features, leading to suboptimal adaptation performance.

In light of the above findings, we introduce CoSDA, a new **Co**ntinual **S**ource-free **D**omain **A**daptation approach that reduces forgetting on all encountered domains and keeps adaptation performance on new domains through teacher-student consistency learning. CoSDA employs a dual-speed optimized teacher-student model pair: a slowly-changing teacher model to retain previous domain knowledge and a fast optimized student model to transfer to new domain. During adaptation, the teacher model infers on target domain to obtain knowledge that matches previous domain features, and the student model learns from this knowledge with consistency loss. We also incorporate mutual information loss to enhance the transferability and robustness to hard domains. Extensive experiments show that CoSDA significantly outperforms other SFDA methods in terms of forgetting index. Moreover, CoSDA does not require prior knowledge such as domain ID and is highly robust to hard domains. CoSDA is easy to implement and can be integrated with other SFDA methods to alleviate forgetting.

## 2 Preliminaries and Related Works

**Preliminaries.** Let $\mathbb{D}_S$ and $\mathbb{D}_T$ denote the source domain and target domain. In domain adaptation, we have one fully-labeled source domain $\mathbb{D}_S$ and $K$ unlabeled target domains $\{\mathbb{D}_{T_k}\}_{k=1}^K$. To ensure confidentiality, all data computations are required to remain local and only the global model $h$ is accessible, which is commonly referred to as source-free domain adaptation (Li et al., 2020; Liang et al., 2020). With this setting, continual DA starts from training an off-the-shelf model $h$ on the source domain, and subsequently transfer it to all target domains. The goal of continual DA is to sustain the model's performance on all previous domains after adaptation. We summarize two adaptation scenarios based on the number of target domains, as depicted in Figure 1I:

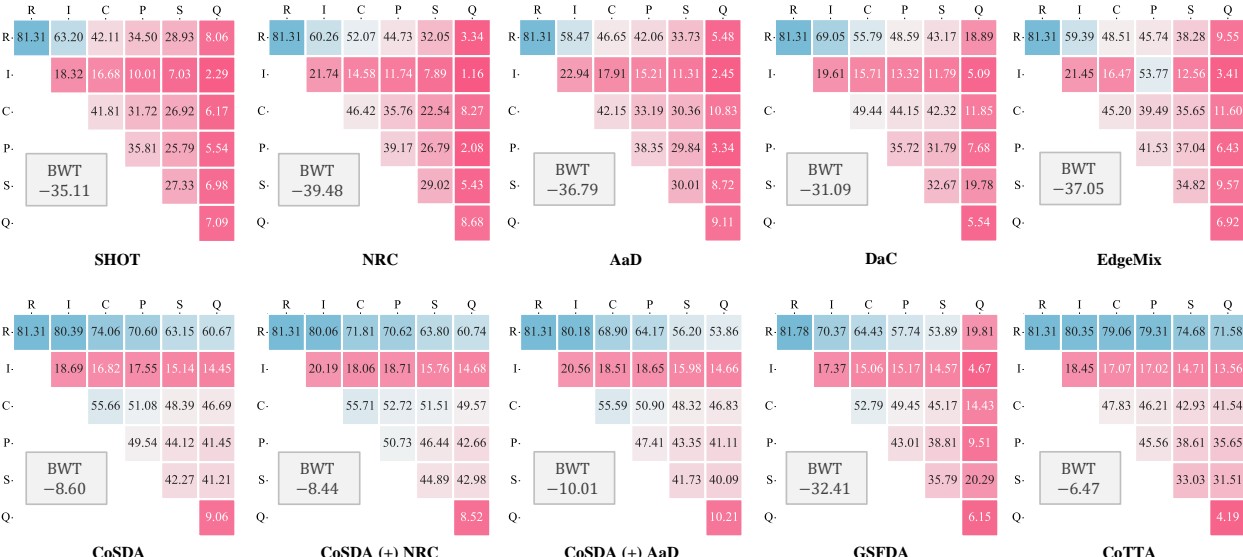

Figure 2: Multi-target sequential adaptation on the DomainNet with the adaptation order of **R**eal→ **I**nfograph→ **C**lipart→ **P**ainting→ **S**ketch→ **Q**uickdraw. The accuracy matrix measures the transferability, with the value in position $(i, j)$ denotes the accuracy on the i-th domain after adaptation on the j-th domain. Backward transfer (BWT) measures the total degree of forgetting with range $-100$ to $0$, where a larger BWT value indicates a smaller degree of forgetting and $0$ indicates no forgetting.

*Single-target adaptation.* We start from $K = 1$, which is most common for current SFDA studies. In this setting, A source pre-trained model is transferred to one target domain and test data is arbitrarily sampled from both source and target domain without prior knowledge such as domain ID.

*Multi-target sequential adaptation.* We extend to $K \geq 2$, where the model is sequentially transferred to each target domain and test data is drawn from all seen domains.

**Related Works.** Current SFDA methods adopt self-training techniques to address domain shift as follows: SHOT (Liang et al., 2020) uses entropy regularization for adaptation; NRC (Yang et al., 2021b) and AaD (Yang et al., 2022) generate pseudo-labels with nearest-neighbor; DaC (Zhang et al., 2022) and EdgeMix (Kundu et al., 2022) adopt data augmentation as consistency loss and SHOT++ (Liang et al., 2022b) designs auxiliary tasks to learn domain-generalized features. Despite the above methods, we further survey two types of methods closely related to CoSDA: knowledge distillation-based methods and continual DA.

*Knowledge distillation-based methods.* Knowledge distillation (Hinton et al., 2015), which transfers knowledge from a well-trained teacher model to a student model, has been widely used in domain adaptation. To enhance adaptation performance, bi-directional distillation is applied in TSML (Li et al., 2023) while SSNLL (Chen et al., 2022) utilizes the mean-teacher (Tarvainen & Valpola, 2017) structure. DINE (Liang et al., 2022a) introduces a memory-bank to store historical inference results, providing better pseudo-labels for the student model. However, in contrast to the dual-speed optimization strategy used in CoSDA, these distillation-based methods update both the teacher and student models simultaneously, leading to the forgetting of previous domain features.

*Continual DA.* A few works have explored continual domain adaptation by incorporating continual learning techniques, which can be summarized into three categories: feature replay (Bobu et al., 2018), dynamic architecture (Mallya & Lazebnik, 2018; Mancini et al., 2019; Yang et al., 2021a) and parameter regularizations (Niu et al., 2022; Wang et al., 2022). CUA (Bobu et al., 2018) and ConDA (Taufique et al., 2021) samples a subset from each target domain as replay data. PackNet (Mallya & Lazebnik, 2018) separates a subset neurons for each task. Aadgraph (Mancini et al., 2019) encodes the connection of previous domains into one dynamic graph and uses it to select features for new domain. GSFDA (Yang et al., 2021a) assigns specific feature masks to different domains. EATA (Niu et al., 2022) uses the elastic-weight consolidation (EWC) (Kirkpatrick et al., 2017) as the regularization loss. CoTTA (Wang et al., 2022) ensures knowledge preservation by stochastically

preserving a subset of the source model's parameters during each update. Distinct from the above methods, CoSDA adopts a dual-speed optimized teacher-student model pair, inspired by LSTM (Hochreiter & Schmidhuber, 1997), to mitigate forgetting. Specifically, a slowly-changing teacher model is utilized to preserve long-term features, while a fast optimized student model is employed to learn domain-specific features.

# 3 CoSDA: An Approach for Continual SFDA

**Overview.** CoSDA is a continual source-free domain adaptation method that achieves multi-target sequential adaptation through pseudo-label learning. For continual learning, CoSDA uses the features learned from previous domains to construct pseudo-labels, which are then used for both adapting to new target domains and preventing forgetting on previously encountered domains. Inspired by knowledge distillation (Hinton et al., 2015), CoSDA utilizes a dual-speed optimized teacher-student model pair, consisting of the teacher model $h_{\boldsymbol{\theta}}$ which retains the knowledge of previous domains, and the student model $h_{\boldsymbol{\psi}}$ that learns domain-specific features. The teacher model generates pseudo-labels for the student model during training, and the student model learns from both the target data and the pseudo-labels using a consistency loss. After adaptation, the teacher model serves as the global model. The framework of CoSDA is presented in Figure 1II, and the details are discussed below.

## 3.1 Consistency Learning with Teacher Knowledge

For each data point $\mathbf{X}$ from current target domain $\mathbb{D}_{T_k}$, we obtain the classification score from the teacher model $h_{\boldsymbol{\theta}}(\mathbf{X})$, and use it as the pseudo-label to train the student model. However, directly learning from $h_{\boldsymbol{\theta}}(\mathbf{X})$ may lead to overfitting to the teacher model. To address this issue, we introduce a consistency loss that consists of three steps. First, we compress the soft-label $h_{\boldsymbol{\theta}}(\mathbf{X})$ into a hard-label $\mathbf{p}$ with a temperature parameter $\tau$ as $\mathbf{p} := \mathrm{softmax}\left(h_{\boldsymbol{\theta}}(\mathbf{X})/\tau\right)$. Next, we augment $\mathbf{X}$ and train the student model to be consistent with the hard-label $\mathbf{p}$ for the augmented samples. Among existing methods (Chen et al., 2020; Cubuk et al., 2019), We choose mixup (Zhang et al., 2018) as the augmentation strategy for three advantages: (1) Mixup has the lowest computation cost. Non-mixup augmentation methods typically require $k\times$ data-augmentations and model inferences for each sample (e.g., $k = 4$ for MixMatch (Berthelot et al., 2019) and 32 for CoTTA (Wang et al., 2022)), while mixup works with $k = 1$ and therefore does not require any extra computations. (2) Mixup can be applied to other data modalities, such as NLP (Guo et al., 2019) and Audio (Meng et al., 2021), while other methods are specifically designed for image data. (3) Mixup facilitates the learning of domain-invariant features. Recent studies (Carratino et al., 2022; Zhang et al., 2021) point out that mixup can contract the data points towards their domain centroid, thereby holistically reducing the domain distance (details are provided in Appendix A.1). With mixup augmentation, we construct the consistency loss as follows:

*Consistency learning with mixup.* For a random-sampled data pair $(\mathbf{X}_i, \mathbf{X}_j)$ with hard-labels $(\mathbf{p}_i, \mathbf{p}_j)$. We sample $\lambda \sim \mathrm{Beta}(a, a)$ and construct the mixed data point as

$$\tilde{\mathbf{X}} = \lambda \mathbf{X}_i + (1 - \lambda)\mathbf{X}_j; \quad \tilde{\mathbf{p}} = \lambda \mathbf{p}_i + (1 - \lambda)\mathbf{p}_j, \tag{1}$$

then the consistency loss for $h_{\boldsymbol{\psi}}$ is

$$\ell_{\mathrm{cons}}(\tilde{\mathbf{X}}, \tilde{\mathbf{p}}; \boldsymbol{\psi}) := D_{\mathrm{KL}}\left(\tilde{\mathbf{p}} \| h_{\boldsymbol{\psi}}(\tilde{\mathbf{X}})\right). \tag{2}$$

Consistency loss helps student model to learn from both previous domain knowedge and the target domain features. However, when the target data is extremely different from the previous domains, the consistency loss may cause the model collapse. To improve the robustness of the model and enable it to learn from hard domains, we employ the mutual information (MI) loss as the regularization:

*Mutual information maximization.* For a batch of mixed data $\{\tilde{\mathbf{X}}_i\}_{i=1}^{B}$, we obtain the marginal inference results as $\bar{\mathbf{h}}_{\boldsymbol{\psi}} = \frac{1}{B}\sum_{i=1}^{B} h_{\boldsymbol{\psi}}(\tilde{\mathbf{X}}_i)$ and formalize the MI as follows:

$$\mathrm{MI}\left(\{h_{\boldsymbol{\psi}}(\tilde{\mathbf{X}}_i)\}_{i=1}^{B}\right) := -\frac{1}{B}\sum_{i=1}^{B} D_{\mathrm{KL}}\left(h_{\boldsymbol{\psi}}(\tilde{\mathbf{X}}_i) \| \bar{\mathbf{h}}_{\boldsymbol{\psi}}\right). \tag{3}$$

---

**Algorithm 1** Adaptation process of CoSDA for $\mathbb{D}_{T_k}$

---

1: **Inputs:** global model $h$, unlabeled training set $\mathbb{D}_{T_k}$.
2: **Hypars:** total epochs $E$, learning rate $\eta$, temperature $\tau$, mixup Beta$(a, a)$, loss weight $\alpha$, EMA momentum $m$.
3: Initialize $\boldsymbol{\theta} \leftarrow h, \boldsymbol{\psi} \leftarrow h, (\boldsymbol{\mu}, \mathbf{Var}) \leftarrow h$;
4: **for** $t = 0$ **to** $E-1$ **do**
5:     **for** every mini-batch $\mathbf{X}$ in $\mathbb{D}_{T_k}$ **do**
6:         Init. $\mathbf{p} = \text{softmax}(h_{\boldsymbol{\theta}}(\mathbf{X})/\tau)$, $\lambda \sim \text{Beta}(a, a)$;
7:         Mixup. $(\tilde{\mathbf{X}}, \tilde{\mathbf{p}}) = \lambda(\mathbf{X}, \mathbf{p}) + (1 - \lambda)\text{Shuffle}(\mathbf{X}, \mathbf{p})$;
8:         Infer. $\ell(\tilde{\mathbf{X}}, \tilde{\mathbf{p}}; \boldsymbol{\psi}) = \ell_{\text{cons}}(\tilde{\mathbf{X}}, \tilde{\mathbf{p}}; \boldsymbol{\psi}) + \alpha\ell_{\text{MI}}(\tilde{\mathbf{X}}; \boldsymbol{\psi})$;
9:         SGD. $\boldsymbol{\psi} \leftarrow \boldsymbol{\psi} - \eta \cdot \nabla_{\boldsymbol{\psi}}\ell(\tilde{\mathbf{X}}, \tilde{\mathbf{p}}; \boldsymbol{\psi})$;                                 *# Student*
10:     **end for**
11:     EMA. $\boldsymbol{\theta} \leftarrow m \cdot \boldsymbol{\theta} + (1 - m) \cdot \boldsymbol{\psi}$;                                       *# Teacher*
12:     EMA. $\boldsymbol{\mu} \leftarrow m \cdot \boldsymbol{\mu} + (1 - m) \cdot \boldsymbol{\mu}_{\boldsymbol{\psi}}$;
13:     EMA. $\mathbf{Var} \leftarrow m \cdot \mathbf{Var} + (1 - m) \cdot \mathbf{Var}_{\boldsymbol{\psi}}$.
14: **end for**
15: **Return:** new global model $h$ with params $(\boldsymbol{\theta}, \boldsymbol{\mu}, \mathbf{Var})$.

---

Our goal is to maximize mutual information during training, which is achieved through the related MI loss as $\ell_{\text{MI}} := -\text{MI}(\cdot)$. Based on previous studies (Hu et al., 2017; Liang et al., 2020), $\ell_{\text{MI}}$ can be decomposed into two components: maximizing the instance entropy and minimizing the marginal inference entropy. The former encourages the model to learn distinct semantics for each data sample, while the latter prevents the model from overfitting to only a few classes (see Appendix A.2 for detailed analysis). Experimental results demonstrate that using the MI loss enables CoSDA to adapt to hard domains (such as Quickdraw on DomainNet) without experiencing catastrophic forgetting. The total loss is obtained by combining the consistency loss and MI loss, i.e., $\ell_{\boldsymbol{\psi}} = \ell_{\text{cons}} + \alpha \cdot \ell_{\text{MI}}$.

### 3.2 Dual-Speed Optimization Strategy

In continual domain adaptation, the global model adapts to each target domain in sequence. To prevent forgetting of previously learned features, we are inspired by LSTM for sequence data processing and adopt a dual-speed strategy to optimize the student and teacher models separately, with the student learning short-term features specific to the current domain and the teacher filtering out long-term domain-invariant features. Specifically, the student model is updated rapidly using SGD with loss $\ell_{\boldsymbol{\psi}}$ after every batch, while the teacher model is slowly updated by performing exponential moving average (EMA) between the previous-step teacher model and the current-step student model at the end of each epoch, as depicted in Figure 1II. This dual-speed strategy allows for a smooth knowledge transition between the two models, preventing abrupt changes during adaptation and maintaining the model's performance on previous domains.

*Updating the mean and variance in BatchNorm.* BatchNorm is a widely-used normalization technique in deep learning models, which estimates the mean and variance of the overall dataset as $(\boldsymbol{\mu}, \mathbf{Var})$ and utilizes these statistics to normalize the data. As the $\boldsymbol{\mu}$-$\mathbf{Var}$ statistics can exhibit significant variation across different domains, prior DA methods, such as FedBN (Li et al., 2021) and DSBN (Chang et al., 2019), typically assign distinct statistics to different domains. However, these methods are not applicable to continual DA since the test data randomly comes from all previously encountered domains without prior knowledge of the domain ID. To unify the BN statistics among different domains, we propose a dual-speed updating method for the mean and variance values. During the training process, the student model estimates the mean and variance of the target domain data as $\boldsymbol{\mu}_{\boldsymbol{\psi}}$ and $\mathbf{Var}_{\boldsymbol{\psi}}$ respectively. After each epoch, the teacher model updates its BN statistics using the EMA method as:

$$\boldsymbol{\mu} \leftarrow m\boldsymbol{\mu} + (1 - m)\boldsymbol{\mu}_{\boldsymbol{\psi}}; \ \mathbf{Var} \leftarrow m\mathbf{Var} + (1 - m)\mathbf{Var}_{\boldsymbol{\psi}}. \tag{4}$$

During testing, the teacher model applies the global $(\boldsymbol{\mu}, \mathbf{Var})$ parameters to BatchNorm layers.

### 3.3 Algorithm and Hyper-Parameters

Based on the concepts of consistency learning and dual-speed optimization, we present the operating flow of our CoSDA method in Algorithm 1 as follows: at first, we initialize the teacher and student models with the global model that has been trained on previous domains. During each epoch, we employ consistency learning to train the student model while keeping the teacher model frozen. When an epoch is finished, we use EMA to update the teacher model as well as the mean and variance statistics of BatchNorm. After adaptation, the teacher model serves as the new global model.

CoSDA is easy to *integrate with other SFDA methods* to further mitigate the forgetting. As outlined in Section 3.1, the pseudo-labels for the student model are simply generated by compressing the soft-label from the teacher model. The quality of these pseudo-labels can be further enhanced with advanced SFDA methods such as the memory bank (Yang et al., 2021b; Liang et al., 2022a), kNN (Yang et al., 2022), and graph clustering (Yang et al., 2020). By further refining the inference results from the teacher model, these pseudo-label-based methods can be seamlessly integrated with CoSDA. The results on both single-target (Figures 2,3) and multi-target sequential adaptation (Table 1,2, and 3) extensively show that the integration of CoSDA significantly reduces forgetting while maintaining adaptation performance.

*Implementation details of CoSDA.* We introduce four hyper-parameters: label compression temperature ($\tau$), mixup distribution ($a$), loss weight ($\alpha$) and EMA momentum ($m$). Following prior research on knowledge distillation and mixup (Berthelot et al., 2019), we fix $\tau = 0.07$ and $a = 2$ for all experiments. Our findings suggest that the mutual information (MI) loss function performs well on datasets with a small number of well-defined classes and clear class boundaries, but it may lead to incorrect classification on datasets with a large number of classes exhibiting semantic similarity. Therefore, we set $\alpha$ empirically to 1 for OfficeHome, Office31 and VisDA, and 0.1 for DomainNet. To apply the EMA strategy, we follow the settings in MoCo (He et al., 2020) and BYOL (Grill et al., 2020) and increase the momemtum from 0.9 to 0.99 using a cosine schedule as: $m_t = 0.99 - 0.1 \times \left[\cos\left(\frac{t}{E}\pi\right) + 1\right] / 2$.

## 4 Experiments

We investigate the mechanisms of catastrophic forgetting through a systematic analysis of various continual DA scenarios. First, we conduct extensive experiments on representative methods from SFDA and continual DA, and report their forgetting on several benchmarks. Then we demonstrate the effectiveness of CoSDA in reducing forgetting under both single and multi-target sequential adaptation scenarios. We also analyze the robustness of CoSDA to hard domains. To ensure fairness in comparison, we reimplement the selected methods in a unified framework. **The code** used to reproduce our results is provided as supplementary materials.

### 4.1 Realistic Evaluation of Current Methods

To avoid unfair comparisons that can arise from variations in the backbones, pretraining strategies, total benchmark datasets, etc., we implemented several representative SFDA methods in a unified framework and evaluated them on four benchmarks: DomainNet (Peng et al., 2019), OfficeHome (Venkateswara et al., 2017), Office31 (Saenko et al., 2010), and VisDA (Peng et al., 2017). In detail, we employ the ImageNet-pretained ResNet with a weight-normed feature bottleneck (Liang et al., 2022b) as the backbone, utilize the dual-lr pre-training strategy proposed in SHOT (Liang et al., 2020), and adopt mini-batch SGD with momentum 0.9 as the optimizer. The total number of epochs is set to 20 and batch size is 64. For model-specific hyperparameters, please refer to Appendix A.4. Without loss of generality, we selected six representative methods: (1) SHOT (Liang et al., 2020) and SHOT++ (Liang et al., 2022b) as they are the first to propose the SFDA setting and have been followed by many works such as DINE (Liang et al., 2022a) and Decision (Ahmed et al., 2021). (2) NRC (Yang et al., 2021b) and AaD (Yang et al., 2022) as they perform the best on all benchmarks and can integrate with CoSDA. (3) DaC (Zhang et al., 2022) and Edgemix (Kundu et al., 2022) as they both use data augmentations to construct consistency loss for adaptation, which is similar to our approach. For comparison, we consider two well-performed continual DA methods: GSFDA (Yang et al., 2021a) and CoTTA (Wang et al., 2022).

Table 1: Single target adaptation on DomainNet with six domains. **Vanilla** refers to the accuracy before adaptation. **(-) MI** refers to removing the mutual information loss from CoSDA. **(+) NRC** and **(+) AaD** refer to integrating CoSDA with NRC and AaD. Results are reported as: Adaptation accuracy (upper) and Accuracy drop (lower).

| DA | Vanilla | SHOT | SHOT++ | NRC | AaD | DaC | EdgeMix | GSFDA | CoTTA | CoSDA | CoSDA (-) MI | CoSDA (+) NRC | CoSDA (+) AaD |
|---|---|---|---|---|---|---|---|---|---|---|---|---|---|
| I2R | 44.99 | 59.89 | 60.14 | 59.64 | **60.19** | 54.86 | 60.07 | 52.12 | 48.00 | 55.75 | 55.64 | 59.19 | 58.21 |
| | | 13.69 | 15.95 | 14.44 | 14.18 | 9.21 | 12.87 | 7.50 | **0.64** | 3.27 | 4.95 | 4.34 | 2.73 |
| C2R | 50.83 | 62.80 | 62.47 | 64.14 | 63.20 | 58.83 | **64.70** | 55.26 | 53.60 | 60.31 | 59.92 | 61.29 | 61.04 |
| | | 18.03 | 17.51 | 20.40 | 23.33 | 13.47 | 19.06 | 2.59 | **0.87** | 4.16 | 6.14 | 4.13 | 3.66 |
| P2R | 57.20 | 62.94 | 62.62 | 64.50 | 64.15 | 60.97 | **65.36** | 58.18 | 58.24 | 61.56 | 61.37 | 63.21 | 62.84 |
| | | 14.76 | 13.68 | 14.22 | 17.94 | 6.91 | 15.67 | 2.78 | **0.26** | 1.22 | 3.46 | 3.37 | 3.21 |
| S2R | 45.41 | 61.84 | 62.32 | 62.83 | 62.29 | 59.26 | **63.64** | 56.25 | 51.56 | 60.62 | 60.22 | 60.32 | 60.11 |
| | | 21.43 | 19.03 | 15.66 | 25.61 | 9.95 | 22.00 | 6.12 | **0.33** | 4.42 | 7.02 | 5.54 | 4.38 |
| Q2R | 5.17 | 33.55 | 34.62 | 35.59 | **38.74** | 25.76 | 35.19 | 20.53 | 14.09 | 19.93 | — | 24.58 | 24.99 |
| | | 55.87 | 58.96 | 60.46 | 62.29 | 34.91 | 55.61 | 25.29 | **14.13** | 19.54 | | 20.69 | 16.44 |
| Avg. | 40.72 | 56.20 | 56.43 | 57.34 | 57.71 | 51.94 | **57.79** | 48.47 | 45.10 | 51.63 | — | 53.72 | 53.44 |
| | | 24.76 | 25.03 | 25.04 | 28.67 | 14.89 | 25.04 | 8.86 | **3.25** | 6.52 | | 7.61 | 6.08 |
| R2I | 17.36 | 18.30 | 18.72 | 21.74 | **23.06** | 19.61 | 21.45 | 17.37 | 18.41 | 18.68 | 19.77 | 20.18 | 20.55 |
| | | 18.11 | 18.46 | 21.05 | 22.91 | 12.32 | 21.92 | 11.41 | 2.60 | **0.38** | 2.22 | 1.25 | 1.13 |
| C2I | 14.59 | 15.51 | 15.97 | 17.88 | 19.92 | 15.31 | 17.94 | 14.21 | 15.87 | 16.73 | — | **20.78** | 20.44 |
| | | 22.47 | 21.71 | 31.70 | 24.24 | 23.78 | 26.29 | 8.99 | 2.08 | 3.74 | | 6.24 | 7.29 |
| P2I | 15.23 | 16.86 | 16.64 | 18.28 | 19.65 | 17.18 | 18.46 | 15.84 | 16.33 | 17.76 | 17.87 | 20.27 | **20.65** |
| | | 17.21 | 18.06 | 21.60 | 20.31 | 15.17 | 22.25 | 12.47 | 2.50 | 2.82 | **2.01** | 5.73 | 8.51 |
| S2I | 11.86 | 16.72 | 16.53 | 18.89 | 20.04 | 16.69 | 18.43 | 14.66 | 14.70 | 12.94 | 15.18 | 20.22 | **20.33** |
| | | 20.81 | 23.77 | 16.14 | 17.29 | 18.12 | 20.43 | 13.66 | 3.05 | **2.62** | 3.06 | 9.39 | 9.60 |
| Q2I | 1.09 | 3.38 | 4.47 | 5.98 | 6.96 | 4.03 | **7.18** | 3.03 | 2.54 | 2.05 | — | 5.66 | 5.43 |
| | | 46.04 | 55.57 | 43.99 | 48.47 | 49.38 | 56.35 | 50.07 | 19.73 | **4.24** | | 13.34 | 10.27 |
| Avg. | 12.03 | 14.15 | 14.47 | 16.55 | **17.93** | 14.56 | 16.69 | 13.02 | 13.57 | 13.63 | — | 17.42 | 17.48 |
| | | 24.93 | 27.51 | 26.90 | 26.64 | 23.75 | 29.45 | 19.32 | 5.99 | **2.76** | | 7.19 | 7.36 |
| R2C | 45.62 | 54.82 | 56.09 | 56.42 | 57.54 | 52.12 | **57.66** | 49.53 | 48.26 | 56.37 | 55.83 | 56.64 | 56.85 |
| | | 21.71 | 21.17 | 20.64 | 26.62 | 12.52 | 21.23 | 6.97 | **1.13** | 6.29 | 6.11 | 6.92 | 7.98 |
| I2C | 30.46 | 45.67 | 46.32 | 44.13 | **47.12** | 39.35 | 46.06 | 32.81 | 33.12 | 41.57 | 40.97 | 45.65 | 46.42 |
| | | 14.82 | 14.80 | 18.01 | 13.43 | 10.40 | 13.79 | 8.00 | **1.14** | 6.02 | 8.07 | 6.68 | 7.13 |
| P2C | 40.74 | 50.84 | 50.77 | 51.82 | **53.40** | 45.61 | 52.87 | 44.19 | 43.92 | 50.88 | 50.49 | 52.73 | 52.05 |
| | | 23.33 | 22.41 | 22.39 | 22.91 | 14.48 | 21.50 | 9.61 | **2.23** | 7.66 | 6.53 | 7.53 | 7.62 |
| S2C | 47.48 | 57.26 | 58.19 | 58.93 | 60.28 | 55.79 | 59.35 | 54.06 | 52.00 | **61.28** | 60.60 | 59.24 | 60.22 |
| | | 15.98 | 14.72 | 14.16 | 15.51 | 9.48 | 12.66 | 5.73 | **0.09** | 5.49 | 5.32 | 5.44 | 6.10 |
| Q2C | 10.13 | 33.67 | 35.92 | 38.44 | **39.19** | 31.54 | 38.61 | 26.94 | 23.20 | 32.96 | 32.88 | 34.13 | 34.55 |
| | | 34.80 | 38.95 | 44.27 | 43.13 | 30.98 | 43.81 | 20.68 | **8.98** | 14.58 | 26.88 | 14.39 | 15.07 |
| Avg. | 34.89 | 48.45 | 49.46 | 49.95 | **51.51** | 44.88 | 50.91 | 41.51 | 40.10 | 48.61 | 48.15 | 49.68 | 50.02 |
| | | 22.13 | 22.41 | 23.89 | 24.32 | 15.57 | 22.60 | 10.20 | **2.71** | 8.01 | 10.58 | 8.19 | 8.78 |
| R2P | 45.54 | 50.74 | 50.65 | 52.94 | 53.34 | 50.94 | 53.05 | 49.22 | 47.50 | 54.29 | 53.78 | **54.44** | 53.96 |
| | | 7.73 | 7.77 | 14.06 | 19.85 | 5.46 | 12.68 | 3.71 | **0.35** | 4.47 | 5.03 | 4.67 | 6.71 |
| I2P | 29.09 | 41.57 | 42.38 | 41.80 | 44.47 | 39.61 | 43.18 | 35.71 | 32.40 | 42.40 | 42.52 | **44.69** | 44.27 |
| | | 12.34 | 14.75 | 12.99 | 13.42 | 7.15 | 13.90 | 5.72 | **0.74** | 4.61 | 5.30 | 4.86 | 4.76 |
| C2P | 33.13 | 42.92 | 43.64 | 44.72 | 45.09 | 41.32 | 44.68 | 37.92 | 36.44 | 44.12 | 44.39 | 45.27 | **45.52** |
| | | 18.16 | 16.26 | 18.65 | 19.68 | 14.49 | 16.30 | 3.59 | **1.17** | 4.42 | 6.19 | 4.66 | 6.60 |
| S2P | 32.81 | 46.51 | 46.91 | 48.11 | 48.59 | 47.04 | 47.85 | 43.96 | 40.13 | **50.62** | 49.90 | 48.52 | 48.32 |
| | | 14.13 | 11.39 | 13.27 | 12.78 | 9.14 | 13.68 | 5.95 | **0.42** | 4.85 | 5.44 | 6.04 | 6.56 |
| Q2P | 1.79 | 15.27 | 18.55 | 18.87 | **23.52** | 14.48 | 21.04 | 10.03 | 6.53 | 9.47 | — | 14.81 | 16.70 |
| | | 47.94 | 51.85 | 62.80 | 63.75 | 37.73 | 63.47 | 31.70 | 16.29 | 14.32 | | **13.47** | 14.20 |
| Avg. | 28.47 | 39.40 | 40.43 | 41.29 | **43.00** | 38.68 | 41.96 | 35.37 | 32.60 | 40.18 | — | 41.55 | 41.75 |
| | | 20.06 | 20.40 | 24.35 | 25.90 | 14.79 | 24.01 | 10.13 | **3.79** | 6.53 | | 6.74 | 7.77 |
| R2S | 32.42 | 40.04 | 40.96 | 44.19 | 43.87 | 40.52 | 44.37 | 41.70 | 36.44 | 43.35 | 43.77 | **46.07** | 45.15 |
| | | 24.19 | 23.46 | 22.56 | 31.27 | 18.97 | 24.43 | 14.03 | **2.17** | 8.81 | 8.60 | 8.18 | 10.00 |
| I2S | 24.44 | 32.45 | 35.17 | 34.37 | **37.73** | 30.04 | 35.19 | 27.52 | 27.40 | 32.32 | 32.53 | 36.61 | 37.07 |
| | | 18.99 | 19.37 | 17.27 | 16.77 | 14.04 | 17.31 | 16.41 | **1.78** | 6.05 | 11.17 | 5.56 | 6.11 |
| C2S | 38.40 | 43.86 | 44.59 | 46.25 | 47.14 | 41.41 | 45.98 | 41.70 | 40.53 | 46.11 | 46.43 | **47.67** | 47.21 |
| | | 16.25 | 12.95 | 14.30 | 15.94 | 13.10 | 18.61 | 7.53 | **1.39** | 4.05 | 4.93 | 3.95 | 5.72 |
| P2S | 33.89 | 40.07 | 41.14 | 43.64 | 43.39 | 41.05 | 42.93 | 39.04 | 37.45 | 43.29 | 43.81 | **44.86** | 44.38 |
| | | 21.53 | 19.95 | 16.17 | 23.70 | 12.77 | 19.71 | 15.41 | **3.25** | 6.01 | 6.77 | 5.61 | 8.13 |
| Q2S | 8.23 | 23.43 | 24.49 | 29.54 | 27.65 | 22.20 | **30.31** | 14.58 | 13.68 | 17.14 | — | 23.48 | 24.34 |
| | | 35.69 | 35.29 | 47.71 | 48.55 | 27.02 | 45.97 | 43.90 | **7.20** | 10.06 | | 13.77 | 14.81 |
| Avg. | 27.48 | 35.97 | 37.27 | 39.60 | **39.96** | 35.04 | 39.76 | 32.91 | 31.10 | 36.44 | — | 39.74 | 39.63 |
| | | 23.33 | 22.20 | 23.60 | 27.25 | 17.18 | 25.21 | 19.46 | **3.16** | 7.00 | | 7.41 | 8.95 |
| R2Q | 4.54 | 7.08 | 8.32 | 8.14 | **10.81** | 8.73 | 7.62 | 6.15 | 6.40 | 6.01 | — | 7.04 | 8.58 |
| | | 69.33 | 70.96 | 72.13 | 66.83 | 55.82 | 73.35 | 61.61 | 7.49 | **2.95** | | 5.34 | 5.71 |
| I2Q | 2.36 | 4.97 | 5.21 | 4.28 | **6.89** | 3.38 | 4.91 | 2.89 | 3.20 | 2.97 | — | 3.07 | 5.11 |
| | | 28.45 | 31.29 | 31.28 | 30.99 | 28.18 | 31.53 | 27.20 | 8.41 | **1.80** | | 2.99 | 8.96 |
| C2Q | 9.56 | 14.31 | 14.07 | 15.19 | **18.23** | 12.02 | 14.47 | 11.86 | 11.57 | 11.58 | 8.50 | 12.72 | 15.89 |
| | | 57.27 | 62.85 | 62.36 | 52.51 | 50.44 | 64.73 | 65.98 | 6.11 | **2.07** | 7.97 | 2.20 | 8.24 |
| P2Q | 3.40 | 8.14 | 9.52 | 9.19 | **12.10** | 8.16 | 9.61 | 6.50 | 6.19 | 5.12 | — | 5.70 | 8.99 |
| | | 61.03 | 60.04 | 59.95 | 63.10 | 51.93 | 64.70 | 57.15 | 12.41 | 3.77 | | **2.98** | 9.00 |
| S2Q | 11.11 | 14.55 | 14.65 | 15.37 | **18.59** | 14.32 | 15.32 | 14.73 | 12.62 | 12.66 | 10.56 | 16.34 | 17.61 |
| | | 49.63 | 45.13 | 37.25 | 42.09 | 36.74 | 38.15 | 34.21 | 4.59 | **2.61** | 8.66 | 10.57 | 6.84 |
| Avg. | 6.19 | 9.81 | 10.35 | 10.43 | **13.32** | 9.32 | 10.39 | 8.43 | 8.00 | 7.67 | — | 8.97 | 11.24 |
| | | 53.14 | 54.05 | 52.59 | 51.10 | 44.62 | 54.49 | 49.23 | 7.80 | **2.64** | | 4.82 | 7.75 |

Table 2: Single target adaptation on OfficeHome (with **A** (art), **C** (clipart), **P** (product) and **R** (real-world)) and Office31 (with **A** (amazon), **D** (dslr) and **W** (webcam)). Results are reported as: Adaptation accuracy (upper) and Accuracy drop (lower).

| Method | OfficeHome | | | | | | | | | | | | | Office31 | | | | | | |
|---|---|---|---|---|---|---|---|---|---|---|---|---|---|---|---|---|---|---|---|---|
| | C2A | P2A | R2A | A2C | P2C | R2C | A2P | C2P | R2P | A2R | C2R | P2R | Avg. | D2A | W2A | A2D | W2D | A2W | D2W | Avg. |
| Vanilla | 48.98 | 67.09 | 74.57 | 50.68 | 63.14 | 64.15 | 50.89 | 42.27 | 73.26 | 63.82 | 48.66 | 77.90 | 60.45 | 78.11 | 71.82 | 94.34 | 98.59 | 57.29 | 61.06 | 76.87 |
| SHOT | 66.96 | 65.10 | 72.89 | 58.01 | 57.34 | 60.25 | 75.60 | 76.21 | 82.95 | 79.50 | 76.11 | 81.39 | 71.03 | 73.30 | 74.12 | 88.76 | **100.00** | 89.81 | 97.99 | 87.33 |
| | 17.66 | 9.85 | 6.84 | 17.00 | 11.49 | 12.32 | 11.54 | 16.38 | 6.61 | 7.96 | 15.48 | 7.62 | 11.73 | 8.23 | 6.79 | 6.14 | 0.13 | 7.10 | 0.20 | 4.77 |
| SHOT++ | 67.04 | 65.84 | 72.31 | 59.59 | 58.76 | **62.70** | 76.19 | **76.44** | 83.49 | 79.57 | 76.75 | **81.89** | 71.71 | 74.62 | 75.65 | 88.76 | **100.00** | 92.08 | 97.99 | 88.18 |
| | 16.28 | 11.47 | 8.79 | 15.50 | 12.03 | 12.11 | 12.61 | 16.70 | 8.19 | 8.90 | 16.74 | 7.57 | 12.24 | 7.03 | 9.18 | 6.95 | 0.25 | 6.95 | 0.40 | 5.13 |
| NRC | 66.05 | 64.81 | 72.19 | 60.16 | 58.28 | 61.56 | 77.61 | 75.76 | 83.40 | 80.15 | 76.61 | 78.56 | 71.26 | 75.68 | 74.72 | **93.57** | **100.00** | 92.70 | 98.24 | 89.15 |
| | 22.58 | 20.37 | 20.17 | 16.65 | 23.66 | 25.29 | 12.53 | 22.29 | 10.69 | 14.26 | 19.08 | 15.43 | 18.58 | 11.24 | 8.18 | 9.55 | **0.00** | 8.73 | 0.20 | 6.32 |
| AaD | **70.91** | **67.57** | **73.75** | **60.25** | **60.60** | 60.94 | **78.46** | 76.41 | **84.46** | 81.75 | **78.91** | 81.59 | **72.97** | 75.43 | 75.61 | **93.57** | 99.80 | 92.20 | 98.49 | 89.18 |
| | 23.82 | 17.87 | 12.48 | 19.66 | 16.06 | 15.95 | 14.55 | 21.19 | 8.35 | 12.16 | 18.69 | 11.38 | 16.01 | 10.64 | 14.21 | 8.52 | **0.00** | 7.10 | **0.00** | 6.74 |
| DaC | 66.71 | 65.22 | 72.31 | 58.76 | 57.18 | 61.35 | 74.82 | 74.75 | 82.16 | 80.19 | 76.25 | 80.56 | 70.86 | 73.91 | 75.36 | 89.75 | **100.00** | 90.57 | 98.49 | 88.01 |
| | 12.64 | 7.55 | 6.86 | 11.95 | 8.13 | 9.34 | 8.90 | 9.87 | 4.84 | 4.49 | 9.16 | 4.03 | 8.15 | 5.42 | 3.65 | 4.40 | **0.00** | 4.08 | **0.00** | 2.93 |
| EdgeMix | 66.50 | 63.95 | 71.24 | 58.03 | 54.73 | 60.92 | 77.25 | 74.41 | 83.28 | 79.73 | 75.05 | 79.41 | 70.37 | 75.04 | 72.42 | 91.57 | 99.80 | 91.07 | 98.36 | 88.04 |
| | 12.25 | 11.02 | 4.45 | 5.61 | 5.97 | 10.51 | 9.15 | 7.99 | 5.30 | 7.34 | 12.32 | 9.69 | 8.47 | 7.63 | 17.86 | 9.72 | 0.00 | 7.38 | 0.00 | 7.10 |
| GSFDA | 68.97 | 65.55 | 72.39 | 57.04 | 54.27 | 59.66 | 77.18 | 74.54 | 83.98 | 80.47 | 76.47 | 81.71 | 71.02 | 72.17 | 73.30 | 88.55 | **100.00** | 88.43 | **98.97** | 86.90 |
| | 8.52 | 4.66 | 2.80 | 5.03 | 2.18 | 2.80 | 1.77 | 5.52 | 1.77 | 0.83 | 4.67 | 1.55 | 3.51 | 1.00 | 0.63 | 3.19 | 0.13 | 2.48 | **0.00** | 1.24 |
| CoTTA | 53.73 | 53.32 | 64.98 | 50.45 | 45.59 | 51.68 | 67.45 | 63.14 | 77.79 | 74.87 | 62.79 | 74.23 | 61.67 | 66.88 | 65.53 | 86.14 | 99.80 | 85.16 | 97.74 | 83.54 |
| | **1.58** | 1.01 | **0.13** | **0.17** | 0.25 | **0.59** | 0.00 | 0.04 | 0.02 | 1.03 | 0.02 | 0.40 | | **0.00** | 0.38 | **0.64** | **0.00** | 1.77 | **0.00** | **0.46** |
| CoSDA | 67.86 | 64.94 | 73.34 | 58.85 | 54.75 | 61.15 | 75.44 | 74.50 | 82.83 | 79.78 | 75.03 | 80.65 | 70.76 | 74.90 | 74.16 | 86.75 | **100.00** | 89.43 | 98.62 | 87.31 |
| | 4.42 | 2.73 | 2.41 | 2.97 | 3.06 | 2.96 | 1.36 | 4.58 | 2.06 | 0.99 | 4.42 | 1.76 | 2.81 | 3.61 | 2.14 | 1.03 | **0.00** | 1.63 | **0.00** | 1.40 |
| CoSDA (-) MI | 61.19 | 58.43 | 68.97 | 53.33 | 49.31 | 57.14 | 71.16 | 70.13 | 80.78 | 77.14 | 70.23 | 76.70 | 66.21 | 70.96 | 68.90 | 84.34 | 99.80 | 85.79 | 97.74 | 84.59 |
| | 1.76 | **0.81** | 0.82 | 0.58 | 0.97 | 1.33 | 0.29 | 1.69 | 0.59 | **0.00** | 1.35 | 0.65 | 0.90 | 0.40 | **0.13** | 1.17 | **0.00** | 1.42 | **0.00** | 0.52 |
| CoSDA (+) NRC | 67.41 | 65.84 | 71.82 | 58.51 | 53.86 | 58.53 | 77.49 | 75.56 | 83.89 | **81.89** | 75.14 | 81.20 | 70.93 | 75.93 | 74.26 | 91.37 | **100.00** | 91.37 | 98.49 | 88.57 |
| | 8.31 | 7.93 | 6.67 | 4.58 | 10.70 | 7.59 | 2.06 | 5.20 | 3.60 | 2.56 | 7.03 | 2.70 | 5.74 | 2.21 | 3.65 | 2.27 | **0.00** | 2.41 | **0.00** | 1.76 |
| CoSDA (+) AaD | 67.12 | 66.05 | 73.51 | 58.44 | 55.21 | 61.40 | 76.86 | 74.70 | 83.62 | 80.51 | 75.63 | 80.65 | 71.14 | **76.29** | **76.39** | 92.97 | **100.00** | **93.71** | 98.49 | **89.64** |
| | 4.88 | 5.14 | 3.94 | 1.90 | 2.57 | 4.40 | 1.73 | 3.84 | 2.14 | 0.95 | 4.17 | 1.12 | 3.07 | 2.81 | 3.90 | 2.48 | **0.00** | 3.16 | **0.00** | 2.06 |

We report the adaptation performance and forgetting loss of the above methods on both single-target and multi-target sequential adaptation settings:

*For single-target adaptation*, we traverse all domain combinations and report both the adaptation accuracy on the target domain and the accuracy drop on the source domain.

*For multi-target adaptation*, we follow the studies on the domain distances (Peng et al., 2019; Zhang et al., 2019) and select the shortest path for sequential adaptation, i.e., **R**eal → **I**nfograph → **C**lipart → **P**ainting → **S**ketch → **Q**uickdraw for DomainNet and **A**rt → **C**lipart → **P**roduct → **R**eal-world for OfficeHome. Following the continual learning protocols (Lopez-Paz & Ranzato, 2017; Hadsell et al., 2020), we construct an accuracy matrix $\mathbf{R} \in \mathbb{R}^{K \times K}$ over $K$ target domains, where $\mathbf{R}_{i,j}$ is the accuracy on the i-th domain after adaptation on the j-th domain. The accuracy matrix $\mathbf{R}$ is reported to measure the transferability of the features. Moreover, we use backward transfer (BWT) to measure the degree of forgetting, which is calculated as $\text{BWT} = \frac{1}{K-1} \sum_{i=1}^{K-1} \mathbf{R}_{i,K} - \mathbf{R}_{i,i}$. BWT ranges from $-100$ to $0$, with $-100$ indicating the complete forgetting and $0$ indicating no forgetting.

## 4.2 Single-Target Adaptation

Extensive experiments on DomainNet (Table 1), OfficeHome, Office31 (Table 2), and VisDA (Table 3) reveal a widespread trade-off between the adaptation performance and the forgetting for commonly used SFDA methods, with the accuracy gain on the target domain coming at the cost of significant accuracy drop on the source domain. For example, NRC and AaD achieve the best adaptation performance among all benchmarks, but they also suffer from the highest levels of catastrophic forgetting. We also find that consistency learning can alleviate catastrophic forgetting in methods such as DaC and EdgeMix. Specifically, DaC applies both weak and strong augmentations on the target data and establishes a consistency loss to align the features of the augmented data, while EdgeMix employs a pretrained DexiNed (Soria et al., 2020) model to extract edge information and uses these features as domain-invariant features by fusing them into input data using mixup. However, these methods heavily rely on pre-determined data augmentation strategies, which may not generalize well to all domains. For instance, EdgeMix failed to mitigate catastrophic forgetting on DomainNet, and DaC exhibited significant forgetting on the *infograph* and *quickdraw* of DominNet. Compared to these

methods, CoSDA exhibits a significant reduction in forgetting across all adaptation pairs and does not rely on pre-determined data-augs. The experimental results on DomainNet, OfficeHome, and VisDA demonstrate that CoSDA outperforms SHOT, SHOT++, and DaC in most adaptation scenarios, while reducing the average forgetting to approximately $\frac{1}{3}$ on DomainNet. Moreover, as mentioned in Section 3.1, CoSDA can be combined with pseudo-label-based methods to alleviate forgetting. Results on the four benchmarks demonstrate that CoSDA can be used in conjunction with NRC and AaD to reduce their forgetting to approximately $\frac{1}{10}$ to $\frac{1}{3}$ while incurring only a slight decrease in target accuracy (about 1% on average). Furthermore, by incorporating CoSDA, we achieve the best performance on the C,P,S→I, R,I,C→P, and R,C,P→S adaptation pairs of DomainNet, while significantly mitigating forgetting.

*Comparison among continual DA methods.* GSDA and CoTTA reduce the forgetting by restoring the prior domain information: GSFDA assigns specific feature masks to different domains and CoTTA adapts parameter regularization by stochastically preserving a subset of the source model in each update. The experiments reveal some limitations of the above two methods: GSFDA relies on domain ID for each sample during testing, and CoTTA tends to overfit the source domain and learn less plastic features, leading to poor adaptation performance. CoSDA outperforms these methods by obviating the requirement of domain ID and preserving high adaptation performance on the target domain.

Table 3: Single target domain on VisDA. Results are reported on each of the 12 classes separately, and the ablation study of CoSDA is conducted by successively removing the four components of the method.

| Method | Plane | Bcycl | Bus | Car | Horse | Knife | Mcycl | Person | Plant | Sktbrd | Train | Truck | Avg. |
|---|---|---|---|---|---|---|---|---|---|---|---|---|---|
| Vanilla | 62.84 | 16.59 | 58.6 | 64.59 | 66.31 | 3.71 | 80.33 | 29.56 | 65.78 | 24.17 | 89.86 | 12.41 | 47.9 |
| SHOT | 95.58 | 85.64 | 85.44 | 71.22 | 95.84 | 96.19 | 85.85 | **85.65** | 91.25 | 89.00 | 84.47 | 48.41 | 84.55 |
|  | 1.33 | 0.68 | 51.45 | 31.88 | 2.84 | 1.33 | 52.48 | **0.58** | 0.46 | 20.21 | 19.75 | 3.33 | 15.53 |
| NRC | 95.56 | 87.54 | 82.11 | 63.11 | 95.01 | 93.44 | 88.52 | 80.43 | **95.25** | 88.46 | 87.69 | **62.18** | 84.94 |
|  | 0.08 | 0.20 | 27.39 | 34.81 | 11.23 | 0.36 | 16.77 | 7.60 | 1.67 | 9.82 | 34.04 | 13.90 | 13.16 |
| AaD | 95.39 | 86.35 | 82.87 | 69.63 | 95.18 | 95.13 | 89.77 | 82.52 | 91.58 | 90.92 | 90.20 | 57.60 | 85.60 |
|  | 1.30 | 1.17 | 59.10 | **13.58** | 11.51 | 9.84 | 26.62 | 8.28 | 0.71 | 10.53 | 21.97 | 33.29 | 16.49 |
| DaC | 95.78 | 81.93 | 83.69 | **80.20** | 96.83 | 97.06 | 94.10 | 81.40 | 94.77 | **94.21** | 90.82 | 45.28 | **86.34** |
|  | 0.10 | 0.34 | 40.44 | 30.15 | 9.08 | 1.66 | 14.13 | 7.20 | 0.33 | 1.52 | 13.97 | **2.58** | 10.12 |
| EdgeMix | 95.07 | 88.05 | 84.72 | 70.89 | 95.48 | 93.44 | 83.16 | 78.14 | 92.37 | 89.30 | 88.40 | 48.47 | 83.96 |
|  | 0.65 | 0.94 | 28.31 | 31.11 | 7.84 | 0.32 | 33.45 | 14.30 | 0.54 | 7.68 | 24.74 | 16.69 | 13.88 |
| GSFDA | **96.32** | **90.73** | 83.73 | 69.20 | 96.53 | 92.34 | 86.09 | 80.58 | 93.76 | 92.81 | 88.62 | 45.17 | 84.66 |
|  | 0.07 | 2.04 | **-0.24** | 29.30 | 5.05 | **0.00** | 33.99 | 1.07 | 0.19 | 4.28 | 15.44 | 9.79 | 8.41 |
| CoTTA | 94.50 | 60.14 | **92.62** | 71.20 | 96.08 | 38.41 | **96.13** | 81.39 | 94.61 | 85.39 | 84.57 | 30.81 | 77.15 |
|  | 0.42 | 0.38 | 2.72 | 29.78 | 10.49 | 2.16 | 14.59 | 6.51 | **0.05** | **-0.02** | 15.07 | 19.48 | 8.47 |
| CoSDA | 94.99 | 80.69 | 86.99 | 73.41 | 94.75 | 85.97 | 93.58 | 79.72 | 93.11 | 85.75 | 90.25 | 37.71 | 83.08 |
| (+) NRC | 0.24 | 0.18 | 24.44 | 17.19 | 2.18 | 4.24 | **9.08** | 2.59 | 0.08 | 1.26 | **6.68** | 14.86 | **6.92** |
| CoSDA | 95.29 | 83.29 | 82.46 | 68.65 | 95.33 | 90.69 | 91.66 | 80.80 | 93.45 | 85.05 | 89.91 | 54.27 | 84.24 |
| (+) AaD | 0.23 | **0.16** | 31.43 | 25.83 | **1.70** | 1.30 | 18.15 | 1.78 | 0.11 | 9.53 | 8.94 | 9.75 | 9.08 |
| Ablation Study | | | | | | | | | | | | | |
| CoSDA | 95.04 | 86.76 | 86.69 | 75.13 | 95.58 | 90.98 | 91.95 | 82.66 | 93.38 | 88.99 | 90.01 | 51.30 | 85.71 |
|  | 0.19 | 0.65 | 27.59 | 34.61 | 3.11 | 1.55 | 17.91 | 11.50 | 0.46 | 5.40 | 14.30 | 12.84 | 10.84 |
| (-) Teacher | 89.22 | 77.61 | 73.89 | 28.45 | 64.97 | 34.19 | 79.11 | 58.75 | 73.76 | 71.35 | 66.94 | 60.32 | 64.88 |
|  | 0.49 | 95.93 | 86.19 | 98.04 | 89.28 | 96.45 | 93.02 | 99.41 | 87.89 | 96.03 | 94.62 | 87.23 | 85.38 |
| (-) Dual-Speed | 94.55 | 84.72 | 86.09 | 63.01 | 94.11 | 94.84 | 89.04 | 81.53 | 92.19 | 90.18 | 86.64 | 53.44 | 84.19 |
|  | 1.08 | 9.63 | 28.29 | 57.40 | 21.86 | 12.25 | 49.08 | 37.74 | 2.40 | 9.59 | 33.68 | 32.48 | 24.62 |
| (-) Mixup & MI | 93.36 | 55.94 | 85.52 | 74.07 | 94.33 | 61.40 | 95.20 | 80.25 | 92.72 | 75.00 | 86.75 | 34.62 | 77.43 |
|  | 0.06 | 0.43 | 0.70 | 17.23 | 4.69 | 0.06 | 5.35 | 7.23 | 0.06 | -0.04 | 10.79 | 12.13 | 4.89 |
| (-) Mixup | 94.26 | 78.10 | 85.05 | 71.23 | 94.78 | 84.58 | 91.97 | 82.17 | 92.58 | 87.02 | 85.86 | 42.03 | 82.47 |
|  | 0.24 | 1.10 | 10.34 | 25.21 | 8.71 | 0.21 | 16.65 | 9.22 | 0.35 | 0.37 | 19.03 | 13.06 | 8.71 |
| (-) MI | 94.45 | 72.94 | 89.16 | 76.90 | 95.99 | 74.87 | 94.22 | 79.21 | 93.82 | 72.41 | 88.86 | 29.68 | 80.21 |
|  | 0.10 | 1.07 | 25.14 | 39.05 | 3.63 | 0.06 | 17.44 | 8.12 | 0.40 | 0.38 | 6.66 | 8.04 | 9.17 |

*Comparison with Continual Learning Baselines.* As CoSDA can be integrated into existing SFDA methods to alleviate forgetting, we explore its integration impact relative to two established continual learning methodologies, namely Elastic Weight Consolidation (EWC) (Kirkpatrick et al., 2017) and Synaptic Intelligence (SI) (Zenke et al., 2017). We integrates these methods with NRC and AaD, and compare the results. As

illustrated in Table 4, the integration with CoSDA exhibits a significantly reduced average forgetting by approximately 10 while achieving similar results on the target domain. These findings underscore the effectiveness of CoSDA in reducing forgetting, thereby reinforcing its potential utility in SFDA paradigms.

Table 4: Comparison of the integration of typical continual learning methods (EWC, SI) and CoSDA with NRC and AaD on the OfficeHome dataset.

| Method | C2A | P2A | R2A | A2C | P2C | R2C | A2P | C2P | R2P | A2R | C2R | P2R | Avg. |
|---|---|---|---|---|---|---|---|---|---|---|---|---|---|
| NRC (+) EWC | 67.24 | 66.5 | 72.93 | 60.11 | 58.17 | 62.59 | 79.07 | 78.04 | 83.89 | 81.00 | 76.31 | 80.72 | 72.21 |
| | 15.12 | 13.34 | 11.61 | 22.05 | 13.28 | 18.70 | 14.75 | 19.22 | 9.98 | 11.70 | 20.92 | 7.73 | 14.87 |
| NRC (+) SI | 67.33 | 66.05 | 71.98 | 58.92 | 58.79 | 62.20 | 78.44 | 78.64 | 83.49 | 80.31 | 79.50 | 81.59 | 72.27 |
| | 24.14 | 20.32 | 13.52 | 21.35 | 15.91 | 13.58 | 10.34 | 21.31 | 9.78 | 8.12 | 20.14 | 12.06 | 16.23 |
| NRC (+) CoSDA | 67.41 | 65.84 | 71.82 | 58.51 | 53.86 | 58.53 | 77.49 | 75.56 | 83.89 | 81.89 | 75.14 | 81.20 | 70.93 |
| | 8.31 | 7.93 | 6.67 | 4.58 | 10.70 | 7.59 | 2.06 | 5.20 | 3.60 | 2.56 | 7.03 | 2.70 | 5.74 |
| AaD (+) EWC | 69.47 | 67.78 | 73.47 | 61.24 | 60.64 | 63.67 | 79.84 | 78.08 | 85.24 | 82.30 | 78.84 | 80.77 | 73.45 |
| | 24.5 | 21.11 | 13.24 | 19.08 | 16.41 | 15.31 | 11.00 | 18.15 | 7.89 | 8.45 | 19.78 | 10.26 | 15.43 |
| AaD (+) SI | 68.31 | 68.52 | 73.51 | 61.81 | 59.45 | 62.25 | 78.69 | 77.59 | 85.02 | 81.41 | 79.27 | 81.46 | 73.11 |
| | 21.16 | 14.58 | 11.77 | 17.18 | 14.90 | 13.01 | 15.04 | 18.74 | 9.11 | 10.30 | 20.12 | 9.54 | 14.62 |
| AaD (+) CoSDA | 67.12 | 66.05 | 73.51 | 58.44 | 55.21 | 61.40 | 76.86 | 74.70 | 83.62 | 80.51 | 75.63 | 80.65 | 71.14 |
| | 4.88 | 5.14 | 3.94 | 1.90 | 2.57 | 4.40 | 1.73 | 3.84 | 2.14 | 0.95 | 4.17 | 1.12 | 3.07 |

*Robustness to hard domains.* The *infograph* and *quickdraw* in DomainNet are considered hard and typically exhibit low adaptation performance (Peng et al., 2019; Feng et al., 2021). Results in Table 1 show that CoSDA exhibits robust performance on both hard domains, reducing the forgetting from $\geq 23\%$ to 2.76% and from $\geq 44\%$ to 2.64%, respectively. Additionally, by integrating CoSDA, the robustness of NRC and AaD methods is significantly improved.

## 4.3 Multi-Target Sequential Adaptation

We use two metrics to evaluate multi-target sequential adaptation: feature transferability and degree of forgetting. As mentioned in Section 4.1, we utilize the diagonal entries of the accuracy matrix to measure transferability and BWT to measure the degree of forgetting. As shown in Figure 2 and 3, the BWT indices of prior SFDA methods are remarkably low, indicating severe catastrophic forgetting. For instance, the BWT of SHOT, NRC, and AaD in DomainNet are all below $-35$, which corresponds to a continuous decrease in accuracy from 81.31% to $\leq 10\%$ on the *real* domain. As observed in the single-target adaptation, the forgetting in EdgeMix and DaC is alleviated due to the adoption of consistency loss. For example, DaC alleviates forgetting with the BWT value of $-31$ on DomainNet and $-8$ on OfficeHome. Compared to these methods, CoSDA exhibits a significant reduction in forgetting, with BWT values of $-8.6$ on DomainNet and $-2.24$ on OfficeHome.

Furthermore, we find that catastrophic forgetting not only leads to a decrease in accuracy on previous domains but also impairs the model's ability to adapt to new domains. For single-target adaptation, although NRC and AaD suffer from catastrophic forgetting, they still achieve the best performance on the target domain. However, in multi-domain settings, their performance on subsequent domains becomes much lower than CoSDA. By integrating CoSDA with other SFDA methods, we can simultaneously mitigate catastrophic forgetting and enhance the model's transferability to new domains. For example, by integrating CoSDA with NRC, we improve the BWT from $-39.48$ to $-8.44$ on DomainNet, accompanied by an average increase of 12.34% adaptation accuracy on the *clipart*, *painting*, and *sketch*. Similarly, integrating CoSDA with AaD resulted in an increase in BWT from $-36.79$ to $-10.01$ on DomainNet, accompanied by an average improvement of 11.31% in adaptation accuracy.

*Comparison among continual DA methods.* In single domain adaptation (Sec 4.2), we discuss the limitations of GSFDA and CoTTA, with GSFDA relying on domain ID during testing and CoTTA having suffered from limited transferability. These limitations become more severe in multi-domain settings. For instance, GSFDA needs to store features for each domain, leading to a decrease in transferability and difficulty in fitting to hard domains in large-scale datasets with many categories, such as DomainNet. However, in benchmarks with a small number of categories such as OfficeHome,

GSFDA performs well in both transferability and mitigating catastrophic forgetting. CoTTA tends to overfit to the source domain, leading to a continuous drop in performance on the target domain until it becomes unfeasible for transfer. In contrast, CoSDA exhibits superior transferability, surpassing GSFDA by 4.02% on average and CoTTA by 5.23%, and also outperforms GSFDA in terms of BWT.

## 4.4 Ablation Study: Why CoSDA Works

In this section, we perform several ablation studies to investigate the mechanisms underlying CoSDA's transferability and forgetting prevention. We approach this through both quantitative and qualitative analysis, focusing on the adaptation performance and feature visualization. As discussed in Section 3, we design a teacher-student structure as well as a consistency loss to enable adaptation and utilize dual-speed optimization to prevent forgetting. Specifically, we employ mixup to generate pseudo-labels for the consistency loss and introduce MI loss to enhance robustness to hard domains.

First, we conduct domain adaptation on the Office-Home dataset to investigate the impact of momentum values and teacher update frequency on the dual-speed optimization strategy described in Section 3.2. As illustrated in Table 5, a lower momentum value, exemplified by $m = 0.9$, yields an average adaptation accuracy of 71.65, surpassing the 70.76 achieved by CoSDA. However, this setting also leads to more pronounced average forgetting in the source domain, at 5.74, noticeably higher than CoSDA's forgetting rate of 2.81. Conversely, a higher momentum value of $m = 0.99$ significantly reduces forgetting to 0.33,

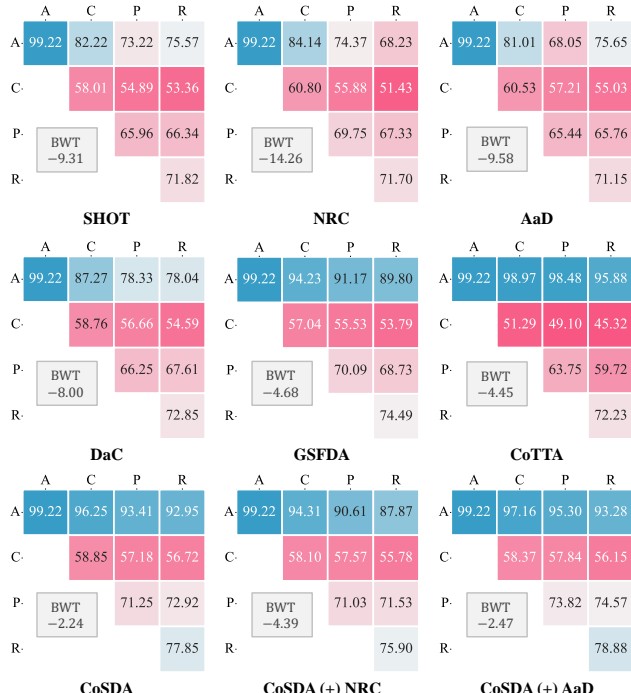

Figure 3: Multi-target sequential adaptation on the OfficeHome with the order of **A**rt→**C**lipart→**P**roduct→**R**eal_world.

but at the cost of reduced adaptation capability, evidenced by a much lower average accuracy of 63.88. These findings underscore the role of momentum in balancing the trade-off between adaptation gain and forgetting loss. In CoSDA, we incrementally adjust the momentum from 0.9 to 0.99 employing a cosine scheduler to optimize this trade-off.

Table 5: Ablation study investigating the impact of momentum values on the dual-speed optimization strategy on the OfficeHome dataset. Specifically, in CoSDA, we increase the momentum from 0.9 to 0.99 using a cosine scheduler.

| Momentum $m$ | C2A | P2A | R2A | A2C | P2C | R2C | A2P | C2P | R2P | A2R | C2R | P2R | Avg. |
|---|---|---|---|---|---|---|---|---|---|---|---|---|---|
| 0.90 | 67.90 | 67.45 | 74.00 | 58.90 | 56.20 | 62.13 | 76.17 | 75.15 | 83.64 | 80.67 | 76.08 | 81.55 | 71.65 |
| | 6.05 | 6.24 | 3.92 | 7.38 | 7.01 | 7.71 | 4.08 | 8.57 | 3.92 | 3.01 | 7.03 | 3.95 | 5.74 |
| 0.99 | 56.00 | 55.58 | 67.10 | 52.76 | 46.55 | 53.77 | 69.00 | 66.41 | 78.76 | 76.02 | 68.69 | 75.97 | 63.88 |
| | 0.87 | 0.32 | 0.25 | 0.25 | 0.19 | 0.34 | 0.21 | 0.44 | 0.16 | 0.17 | 0.67 | 0.14 | 0.33 |
| $m$ increases | 67.86 | 64.94 | 73.34 | 58.85 | 54.75 | 61.15 | 75.44 | 74.50 | 82.83 | 79.78 | 75.03 | 80.65 | 70.76 |
| from 0.9 to 0.99 | 4.42 | 2.73 | 2.41 | 2.97 | 3.06 | 2.96 | 1.36 | 4.58 | 2.06 | 0.99 | 4.42 | 1.76 | 2.81 |

In a similar vein, the update frequency of the teacher model can also impact the trade-off, as shown in Table 6. When the teacher model is updated with a higher frequency, specifically twice per epoch, we observe an increase in the average forgetting within the source domain, escalating from 2.81 to 5.54, while maintaining comparable adaptation accuracy in the target domain. Conversely, adopting a lower update frequency, such as updating the teacher model once every four epochs, significantly mitigates the average forgetting to 0.50. However, this adjustment also results in a decrease in the average adaptation accuracy on the target domain,

Table 6: Ablation study investigating the impact of teacher update frequency on the dual-speed optimization strategy on the OfficeHome dataset. Specifically, In CoSDA, we update the teacher model once per epoch.

| Frequency | C2A | P2A | R2A | A2C | P2C | R2C | A2P | C2P | R2P | A2R | C2R | P2R | Avg. |
|---|---|---|---|---|---|---|---|---|---|---|---|---|---|
| Once / 4 epochs | 60.77 | 61.10 | 70.33 | 54.18 | 51.00 | 55.60 | 72.81 | 70.87 | 81.46 | 78.86 | 71.15 | 78.26 | 67.20 |
| | 0.73 | 0.18 | 0.50 | 0.46 | 0.50 | 0.83 | 0.09 | 0.87 | 0.48 | 0.13 | 0.92 | 0.32 | 0.50 |
| Once / 2 epochs | 65.55 | 63.82 | 72.18 | 56.88 | 53.47 | 58.44 | 74.99 | 73.17 | 82.50 | 79.64 | 73.70 | 79.76 | 68.57 |
| | 2.04 | 1.09 | 1.22 | 1.20 | 1.47 | 1.65 | 0.54 | 2.27 | 0.76 | 0.41 | 1.90 | 0.77 | 1.32 |
| Once / epoch | 67.86 | 64.94 | 73.34 | 58.85 | 54.75 | 61.15 | 75.44 | 74.50 | 82.83 | 79.78 | 75.03 | 80.65 | 70.76 |
| | 4.42 | 2.73 | 2.41 | 2.97 | 3.06 | 2.96 | 1.36 | 4.58 | 2.06 | 0.99 | 4.42 | 1.76 | 2.81 |
| Twice / epoch | 68.40 | 65.51 | 73.47 | 58.14 | 55.37 | 61.28 | 75.38 | 74.30 | 83.55 | 79.60 | 75.19 | 80.77 | 70.02 |
| | 9.26 | 5.43 | 3.49 | 7.09 | 5.10 | 6.13 | 2.79 | 7.95 | 2.68 | 1.69 | 7.97 | 3.68 | 5.54 |

from 70.76 to 67.20. These findings substantiate the efficacy of the proposed dual-speed strategy and illustrate the impact of these two hyperparameters on the performance of CoSDA.

Then, we conduct domain adaptation on VisDA to further validate our claims. As shown in the lower part of Table 3, we investigate the contributions of each part in our method by successively removing the teacher model, dual-speed optimization, mixup, and MI loss. The first row of the table shows that removing the teacher model and using only the student model for predictions leads to overfitting to certain classes and complete failure of adaptation, highlighting the importance of the teacher-student structure. The second row shows that removing dual-speed optimization and simultaneously updating both teacher and student models hardly affects adaptation accuracy, but leads to severe catastrophic forgetting. This highlights the crucial role of dual-speed optimization in preventing forgetting. The next three rows of the table illustrate the results of removing mixup, MI loss, and both mixup and MI loss, and the results indicate that both mixup and MI loss contribute significantly to improving the adaptation performance. We further conduct ablation study of MI loss on DomainNet. The results in Table 1 show that the removal of MI loss leads to training failure on hard domains, highlighting its crucial role in maintaining robustness.

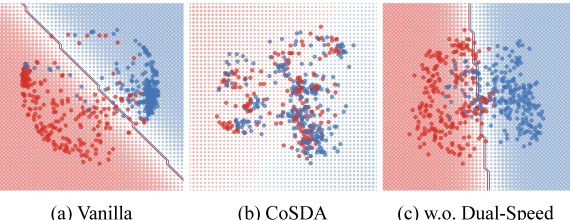

(a) Vanilla     (b) CoSDA     (c) w.o. Dual-Speed

Figure 4: The t-SNE visualizations of the features on VisDA extracted by Vanilla, CoSDA and CoSDA w.o. dual-speed. Red color denotes the source feature and Blue color denotes the target feature. The foreground points denote the data feature, while the background lattice represent the overall feature distributions.

Moreover, we visualize the features of source and target domains under three settings: vanilla, CoSDA, and CoSDA without dual-speed optimization, as shown in Figure 4. Vanilla shows significant distribution shift between source and target domains. After adaptation with CoSDA, we observe that the model learns a shared feature space for both source and target domains, indicating its ability to achieve transferability and prevent catastrophic forgetting. However, without the application of dual-speed optimization, we observe that while some distances between source-target features decrease, others remain distant, suggesting the occurrence of catastrophic forgetting.

## 5 Conclusions and Limitations

In summary, this work conducts a systematic investigation into the issue of catastrophic forgetting on existing domain adaptation methods and introduce a practical DA task named continual source-free domain adaptation. CoSDA, a dual-speed optimized teacher-student consistency learning method, is proposed to mitigate forgetting and enable multi-target sequential adaptation. Extensive evaluations show that CoSDA outperforms state-of-the-art methods with better transferability-stability trade-off, making it a strong baseline for future studies. In addition, our open-source unified implementation framework designed for different SFDA methods can serve as a foundation for further explorations. However, our investigation is currently confined to image classification tasks, further work is needed to extend these findings to other domains.

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

## A   Appendix

### A.1   The Rationale of Selecting Mixup as Data Augmentation Strategy

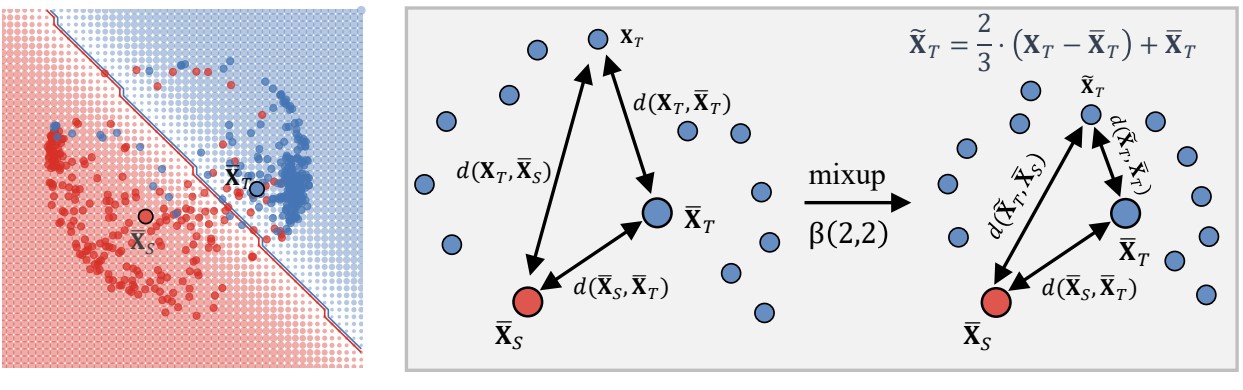

(a) Vanilla feature                                (b) Mixup process

Figure 5: An overview of mixup process on VisDA. Red color denotes the source feature and Blue color denotes the target feature. The centroids of source and target features are denoted by $\bar{\mathbf{X}}_S$ and $\bar{\mathbf{X}}_T$. For CoSDA, we set the value of $\beta(a,a)$ to $a = 2$ and use $\bar{\lambda} = \frac{2}{3}$. The mixup process results in data points shrinking to their centroids, thereby reducing the domain distance.

In Section 3.1, we introduce mixup as a data augmentation strategy used in CoSDA, which is claimed to holistically reduce the domain distance and thereby facilitating the learning of domain-invariant features. In this section, We provide evidence to support this claim. We start with summarizing an equivalent form of mixup proposed by (Carratino et al., 2022) which establishes a connection with label-smoothing techniques as follows:

**Theorem 1.** *Let $\mathbb{D}_T$ be the target domain with $N$ training samples $\mathbf{X}_i$ and their corresponding pseudo-labels $\mathbf{p}_i$. Suppose the mixup augmentation with distribution $\beta_{[0,1]}(a,a)$ are used for the student model $h_{\boldsymbol{\psi}}$ with the consistency loss $\ell_{\mathrm{cons}}(\cdot)$. Then, the empirical risk of the consistency loss can be approximated as:*

$$\xi_{\mathrm{mixup}}(\ell_{\mathrm{cons}}, h_{\boldsymbol{\psi}}) := \frac{1}{N} \sum_{i=1}^{N} \mathbb{E}_{j,\theta} \left[ \ell_{\mathrm{cons}}(\tilde{\mathbf{X}}_i + \boldsymbol{\delta}_i, \tilde{\mathbf{p}}_i + \boldsymbol{\epsilon}_i; \boldsymbol{\psi}) \right], \tag{5}$$

*where $j \sim \mathrm{Unif}(1, \ldots, N)$, $\theta \sim \beta_{[\frac{1}{2}, 1]}(a,a)$, and $(\tilde{\mathbf{X}}_i, \tilde{\mathbf{p}}_i, \boldsymbol{\delta}_i, \boldsymbol{\epsilon}_i)$ can be formulated by squeezing the samples towards their centroid $\bar{\mathbf{X}} = \frac{1}{N} \sum_{i=1}^{N} \mathbf{X}_i$ as follows:*

$$\begin{cases} \tilde{\mathbf{X}}_i = \bar{\theta}(\mathbf{X}_i - \bar{\mathbf{X}}) + \bar{\mathbf{X}}, \\ \tilde{\mathbf{p}}_i = \bar{\theta}(\mathbf{p}_i - \bar{\mathbf{p}}) + \bar{\mathbf{p}}, \\ \boldsymbol{\delta}_i = (\theta - \bar{\theta})\mathbf{X}_i + (1 - \theta)\mathbf{X}_j - (1 - \bar{\theta})\bar{\mathbf{X}}, \\ \boldsymbol{\epsilon}_i = (\theta - \bar{\theta})\mathbf{p}_i + (1 - \theta)\mathbf{p}_j - (1 - \bar{\theta})\bar{\mathbf{p}}, \end{cases} \tag{6}$$

*where $\boldsymbol{\delta}_i, \boldsymbol{\epsilon}_i$ are zero-mean random perturbations, $\|\boldsymbol{\delta}_i\|_2 \ll \|\mathbf{X}_i\|_2$ and $\bar{\theta} = 2 - \frac{a(a-1)}{a-\frac{1}{2}}$ is the expectation of distribution $\theta \sim \beta_{[1/2,1]}(a,a)$. For CoSDA, we have $\bar{\theta} = \frac{2}{3}$ with $a = 2$.*

**Proof.** We recap the format of $\xi_{\mathrm{mixup}}$ as follows:

$$\xi_{\mathrm{mixup}}(\ell_{\mathrm{cons}}, h_{\boldsymbol{\psi}}) := \frac{1}{N^2} \sum_{i=1}^{N} \sum_{j=1}^{N} \mathbb{E}_{\lambda} \left[ \ell_{\mathrm{cons}} \left( h_{\boldsymbol{\psi}}(\lambda \mathbf{X}_i + (1 - \lambda)\mathbf{X}_j, \lambda \mathbf{p}_i + (1 - \lambda)\mathbf{p}_j) \right) \right], \tag{7}$$

where $\lambda \sim \beta_{[0,1]}(a,a)$. To investigate the impact of $\lambda$ on Eq. (7), we construct a function that relates the value of $\lambda$ to mixup data pairs as $m_{i,j}(\lambda)$:

$$m_{i,j}(\lambda) = \ell_{\text{cons}}\left(h_{\boldsymbol{\psi}}(\lambda \mathbf{X}_i + (1-\lambda)\mathbf{X}_j, \lambda \mathbf{p}_i + (1-\lambda)\mathbf{p}_j)\right). \tag{8}$$

Denoting $\lambda = (1-\pi)\theta + \pi\theta', \theta \sim \beta_{[\frac{1}{2},1]}(a,a), \theta' \sim \beta_{[0,\frac{1}{2}]}(a,a), \pi \sim \text{Ber}(\frac{1}{2})$, we can rewrite $m_{i,j}(\lambda)$ as

$$\mathbb{E}_\lambda\left[m_{i,j}(\lambda)\right] = \mathbb{E}_{\theta,\theta',\pi}\left[m_{i,j}\left((1-\pi)\theta + \pi\theta'\right)\right] = \frac{1}{2}\left(\mathbb{E}_\theta\left[m_{i,j}(\theta)\right] + \mathbb{E}_{\theta'}\left[m_{i,j}(\theta')\right]\right). \tag{9}$$

Since $\theta' = 1 - \theta$, we have $\mathbb{E}_\theta\left[m_{i,j}(\theta)\right] = \mathbb{E}_{\theta'}\left[m_{i,j}(\theta')\right]$. Substituting it into Eq. (7), we obtain:

$$\xi_{\text{mixup}}(\ell_{\text{cons}}, h_{\boldsymbol{\psi}}) = \frac{1}{N^2}\sum_{i=1}^{N}\sum_{j=1}^{N}\mathbb{E}_\theta\left[m_{i,j}(\theta)\right] = \frac{1}{N}\sum_{i=1}^{N}\left(\frac{1}{N}\sum_{j=1}^{N}\mathbb{E}_\theta\left[m_{i,j}(\theta)\right]\right). \tag{10}$$

Denote $\xi_i = \frac{1}{N}\sum_{j=1}^{N}\mathbb{E}_\theta\left[m_{i,j}(\theta)\right]$, we have $\xi_i = \mathbb{E}_{\theta,j}\left[\ell_{\text{cons}}\left(h_{\boldsymbol{\psi}}(\theta\mathbf{X}_i + (1-\theta)\mathbf{X}_j), \theta\mathbf{p}_i + (1-\theta)\mathbf{p}_j)\right]\right]$. Notably, $(\tilde{\mathbf{X}}_i, \tilde{\mathbf{p}}_i)$ has the following relation with $\xi_i$:

$$\tilde{\mathbf{X}}_i = \bar{\theta}(\mathbf{X}_i - \bar{\mathbf{X}}) + \bar{\mathbf{X}} = \mathbb{E}_{\theta,j}[\theta\mathbf{X}_i + (1-\theta)\mathbf{X}_j]; \quad \tilde{\mathbf{p}}_i = \bar{\theta}(\mathbf{p}_i - \bar{\mathbf{p}}) + \bar{\mathbf{p}} = \mathbb{E}_{\theta,j}[\theta\mathbf{p}_i + (1-\theta)\mathbf{p}_j]. \tag{11}$$

With the relations in Eq. (11), we denote $\boldsymbol{\epsilon}, \boldsymbol{\delta}$ and prove Eq. (5) as follows

$$\boldsymbol{\delta}_i = \theta\mathbf{X}_i + (1-\theta)\mathbf{X}_j - \tilde{\mathbf{X}}_i; \quad \boldsymbol{\epsilon}_i = \theta\mathbf{p}_i + (1-\theta)\mathbf{p}_j - \tilde{\mathbf{p}}_i; \quad \xi_i = \mathbb{E}_{\theta,j}\left[\ell_{\text{cons}}\left(h_{\boldsymbol{\psi}}(\tilde{\mathbf{X}}_i + \boldsymbol{\delta}_i), \tilde{\mathbf{p}}_i + \boldsymbol{\epsilon}_i)\right)\right], \tag{12}$$

Combining the equations Eq. (11) and Eq. (12), we can obtain $\mathbb{E}[\boldsymbol{\delta}_i] = 0$ and $\mathbb{E}[\boldsymbol{\epsilon}_i] = 0$. Furthermore, following the empirical study in (Carratino et al., 2022), we can conclude that the $\ell_2$-norm of $\boldsymbol{\delta}_i$ is much smaller than that of $\mathbf{X}_i$. $\qquad\square$

**We conduct the following analysis of Theorem 1.** Since the magnitude of the perturbation $\|\boldsymbol{\delta}\|_2$ is much smaller than the norm of the mixed samples $\|\tilde{\mathbf{X}}\|_2$ (Carratino et al., 2022), we can interpret the mixup augmentation as squeezing the samples towards their centroid, i.e., $\mathbf{X} \to \tilde{\mathbf{X}}$. In domain adaptation, the cluster distance between source and target domains (Deng et al., 2019) is often used to measure the degree of distribution shift. As shown in Figure 5, a source-domain-trained model has a clear boundary in the distributions of source and target domains (as shown in (a)). However, the centroids of the source and target domains are much closer than the sample points. By using the mixup method, all sample points are squeezed towards the centroids (as shown in (b)), thereby heuristically reducing the domain distance and facilitating the learning of domain-invariant features.

## A.2 The Analysis of Mutual Information Loss

Mutual information (MI) is a concept used to quantify the degree of dependence between two random variables. It measures the reduction in uncertainty of one variable by knowing the value of the other variable, indicating the amount of information they share. The mutual information between two random variables $\mathbf{X}$ and $\mathbf{y}$ is defined as follows:

$$\text{MI}(\mathbf{X}, \mathbf{y}) = D_{\text{KL}}\left(p(\mathbf{X},\mathbf{y})\|p(\mathbf{X})p(\mathbf{y})\right). \tag{13}$$

During the training process of CoSDA, we use $\mathbf{y}$ to denote the label and use $h_{\boldsymbol{\psi}}(\mathbf{X})$ as the label distribution $p(\mathbf{y}|\mathbf{X})$. For $B$ samples in a mini-batch, we estimate the distribution of data $\mathbf{X}$ using empirical distribution as $p(\mathbf{X}) = \frac{1}{B}$. Then we estimate $p(\mathbf{y})$ as $p(\mathbf{y}) = \sum_{\mathbf{X}} p(\mathbf{y}|\mathbf{X})p(\mathbf{X}) = \frac{1}{B}\sum_{i=1}^{B} h_{\boldsymbol{\psi}}(\mathbf{X}_i) := \bar{\mathbf{h}}_{\boldsymbol{\psi}}$. Based on the definitions above, the mutual information for CoSDA can be expressed as follows:

$$\text{MI}(\{\mathbf{X}_i\}_{i=1}^{B}, \boldsymbol{\psi}) = -\frac{1}{B}\sum_{i=1}^{B} D_{\text{KL}}\left(h_{\boldsymbol{\psi}}(\mathbf{X}_i)\|\bar{\mathbf{h}}_{\boldsymbol{\psi}}\right), \tag{14}$$

and the mutual information loss is $\ell_{\text{MI}} := -\text{MI}(\{\mathbf{X}_i\}_{i=1}^{B}, \boldsymbol{\psi})$.

In Section 3.2, we claim the mutual information loss can improve the robustness of the model and enable it to learn from hard domains. We provide evidence to support this claim. Based on previous studies (Liang et al., 2020; Hu et al., 2017), $\ell_{\text{MI}}$ can be decomposed into two components: minimizing the instance entropy and maximizing the marginal inference entropy:

Table 7: Hyperparameters for all the methods evaluated in the experiments.

| Method | Shared hyperparameters | Dataset specific hyperparameters |
|---|---|---|
| SHOT&SHOT++ | learning rate: $1 \times 10^{-3} \sim 2 \times 10^{-4}$ for backbone; learning rate: $1 \times 10^{-2} \sim 2 \times 10^{-3}$ for bottleneck and classifier; cls_loss weight: 0.3; ent_loss weight: 1.0; ssl_loss weight: 0.6 (for SHOT++). | Same for all datasets. |
| NRC | learning rate: $2 \times 10^{-3} \sim 1 \times 10^{-3}$; $k = 4$, $m = 3$. | For VisDA, set learning rate: $2 \times 10^{-3} \sim 1 \times 10^{-4}$,$k = 8$, $m = 8$; For DomainNet, set $k = 6$, $m = 4$. |
| AaD | learning rate: $2 \times 10^{-3} \sim 1 \times 10^{-3}$; $\alpha = 0.4$, decay $\gamma = 0.96$. | For VisDA, set learning rate: $2 \times 10^{-3} \sim 1 \times 10^{-4}$; $k = 2, 3, 4, 8$ For Office-Home, Office31, DomainNet, VisDA. |
| DaC | learning rate: $2 \times 10^{-3} \sim 2 \times 10^{-4}$; temperature: 0.05, K: 40, k: 5; momentum: 0.8, threshold: 0, gate: 0.97; coefficients for cls, im, con and mmd: (0.02, 0.25, 0.03, 0.15). | For VisDA, set learning rate: $5 \times 10^{-4} \sim 6 \times 10^{-5}$; K: 300, momentum: 0.2; coefficients for cls, im, con and mmd: (0.39, 0.1, 1.0, 0.3) ; For DomainNet, set learning rate: $1 \times 10^{-3} \sim 2 \times 10^{-4}$. |
| EdgeMix | learning rate: $1 \times 10^{-3} \sim 2 \times 10^{-4}$; $\lambda = 0.9$, finetune epochs: 2. | For Office31, learning rate: $2 \times 10^{-3} \sim 1 \times 10^{-3}$; For VisDA, learning rate: $2 \times 10^{-3} \sim 1 \times 10^{-4}$. |
| GSFDA | learning rate: $1 \times 10^{-3} \sim 2 \times 10^{-4}$ for backbone; learning rate: $1 \times 10^{-4} \sim 2 \times 10^{-5}$ for bottleneck and classifier; $k = 2$, $s = 100$, $\lambda_{gen} : 1$. | For VisDA, set $k = 10$; For DomainNet, set backbone learning rate: $5 \times 10^{-4} \sim 1 \times 10^{-5}$; bottleneck and classifier learning rate: $5 \times 10^{-5} \sim 1 \times 10^{-6}$; $k = 10$, $\lambda_{gen} = 2$ . |
| CoTTA | source model preserve ratio (rst): 0.01; average predictive prob (ap): 0.92; aug times: 32. | For Office31and VisDA, set ap to 0.5 and rst to 0.001. |
| CoSDA | learning rate: $2 \times 10^{-3} \sim 1 \times 10^{-3}$; temperature: $\tau = 0.07$; mixup: $\beta(2, 2)$; loss weight $\alpha = 1$; EMA momentum: $m = [0.9, 0.99]$. | For DomainNet, set $\alpha = 0.1$ and $m = [0.95, 0.99]$; For VisDA, set learning rate: $4 \times 10^{-3} \sim 2 \times 10^{-3}$. |

1. Minimize instance entropy:

$$\min_{\boldsymbol{\psi}} \frac{1}{B} \sum_{i=1}^{B} \mathrm{ent}(h_{\boldsymbol{\psi}}(\mathbf{X}_i)) := \min_{\boldsymbol{\psi}} \frac{1}{B} \sum_{i=1}^{B} \sum_{c=1}^{C} -h_{\boldsymbol{\psi}}(\mathbf{X}_i)_{i,c} \log h_{\boldsymbol{\psi}}(\mathbf{X}_i)_{i,c}. \tag{15}$$

2. Maxmizie marginal entropy:

$$\max_{\boldsymbol{\psi}} \mathrm{ent}(\bar{\mathbf{h}}_{\boldsymbol{\psi}}) := \max_{\boldsymbol{\psi}} \sum_{c=1}^{C} -\bar{\mathbf{h}}_{\boldsymbol{\psi},c} \log \bar{\mathbf{h}}_{\boldsymbol{\psi},c}. \tag{16}$$

By minimizing instance entropy, the model learns to assign distinct semantics for each data, resulting in a concentrated classification distribution. This enables the model to learn classification information even in the presence of inaccurate pseudo-labels in hard domains. By maximizing the marginal entropy, we ensure that the model learns a uniform marginal distribution, which allows it to learn information from all classes in a broad and balanced manner, rather than overfitting to a few specific classes. Based on the above two advantages, we demonstrate that integrating mutual information loss into the training objective can lead to good properties such as improved robustness and effective learning from hard domains.

### A.3 Training Paradigm

In our experiments, we follow previous settings (Long et al., 2015; Ganin et al., 2016; Liang et al., 2020; Yang et al., 2021b; Zhang et al., 2022) and utilize two common training paradigms: inductive learning and transductive learning. For DomainNet that provides an official train-test split, we use the inductive learning pipeline to train models on the training set and report model performance on the test set. On the other hand, for OfficeHome, Office31, and VisDA, which do not provide an official train-test split, most methods adopt the transductive learning based adaptation pipeline. In this pipeline, the training and testing datasets are identical. Specifically, models are trained on the entire source domain and adapted to the entire unlabeled target domain. During testing, the models are evaluated on the same training dataset to report the adaptation performance as well as the degree of catastrophic forgetting. It is important to note that, since the inductive

learning paradigm is more practical, we primarily focus on the analysis and discussion of our results based on the DomainNet experiments, supplemented by the transductive learning performance on the other three benchmarks.

## A.4 Hyperparameters

We build a unified implementation for all methods with the following settings: we use ResNet50 as the backbone for DomainNet, OfficeHome, and Office31, and ResNet101 for VisDA. We apply cyclic cosine annealing as the learning rate schedule, set the weight decay to $5 \times 10^{-3}$, and use random horizontal flip as regular data augmentations, except for DaC, EdgeMix, and CoTTA. Specifically, EdgeMix uses a pretrained DexiNed (Soria et al., 2020) to extract and confuse edge features, while DaC and CoTTA use AutoAug (Cubuk et al., 2019) as augmentation strategies.

In addition, we construct a validation dataset to select suitable model-specific hyperparameters. Specifically, we split 5% of the data from the current target domain's training set as the validation dataset. In the CoSDA setting, we are not allowed to access the data from previous domains. Therefore, we do not construct a validation dataset on the source domain or previously encountered target domains. The details of hyperparameter selection for each method are presented in Table 7.

