# OpenReview forum: "CoSDA: Continual Source-Free Domain Adaptation"
_TMLR — Withdrawn by Authors_

### Review · Reviewer_ys4R · 2024-11-29

**Summary Of Contributions:**

This work proposes a dual-speed optimized teacher-student model pair that can maintain the performance on multiple sequential target domains in SFDA by EMA.

**Audience:**

Yes

**Claims And Evidence:**

No

**Requested Changes:**

- Is this Continual Source-Free Domain Adaptation a novel problem setting? If so, it should be highlighted, especially regarding the difference from TTA.
- The author said the model is tested on all previously seen domains. Does it include the source domain? If so, I am curious about the results excluding the source domain.
- On page 4, why is hard label $p$ obtained by softmax?
- The purpose of MI is to balance the prediction for all classes. How does it help to learn from hard domains?
- It would be better to place the accuracy drop on the right side of the adaptation accuracy, as the current version is a bit hard to read.
- In multi-target tasks, which one is the source domain?
- How are other baselines like SHOT trained and tested in multi-target tasks? Instead of training the target data sequentially,  ensembling separated target models may give better results.
- The advantage of the proposed method lies in multi-target tasks, but the explanation of single-target tasks takes too much space. I suggest rewriting the multi-target part by adding some tables since the current figures are confusing, and I do not know where to look.
-  The order in multi-target tasks can affect the final results. I would expect an analysis regarding this.
-  The momentum is crucial in memorizing the previous domain, whose optimal value is related to sequence length. I wonder how this is implemented in sequential multi-target tasks, which can have unknown lengths.
- The t-sne visualization provides little information. It would be better to include a comparison with other baselines, especially regarding multi-target data cases.
- The baselines seem out-of-date, considering this a submission at the end of 2024 [1,2,3].

***
ODS: Test-Time Adaptation in the Presence of Open-World Data Shift, ICML 2023

Test-Time Domain Adaptation by Learning Domain-Aware Batch Normalization, AAAI 2024

A Versatile Framework for Continual Test-Time Domain Adaptation: Balancing Discriminability and Generalizability, CVPR 2024

**Strengths And Weaknesses:**

Strengths:
- Overall, the paper is easy to follow, and the proposed algorithm is simple to implement.


Weaknesses:
- The motivation to preserve source domain information is unclear as the goal in DA is to classify target data.
- The comparisons in multi-target data lack details. Some baselines cannot store previous target models by default, but this can be achieved by ensembling.

---

### Review · Reviewer_hJmv · 2024-12-31

**Summary Of Contributions:**

The paper proposes a unified framework for continual source-free domain adaptation. The authors re-implement previous SFDA approaches and show that there is a trade-off between adaptation gain and forgetting loss. Based on this observation, they propose a novel approach to mitigate forgetting. The proposed approach uses a dual-speed optimization strategy and consistency learning for effective adaptation on target tasks. Furthermore, the proposed method outperforms previous state-of-the-art approaches on four different benchmarks in terms of forgetting index.

**Audience:**

Yes

**Claims And Evidence:**

Yes

**Requested Changes:**

1. The paper claims to have introduced the task of continual source-free domain adaptation however there already exists previous work in the area. One of them is:

Ahmed W, Morerio P, Murino V. Continual source-free unsupervised domain adaptation. InInternational Conference on Image Analysis and Processing 2023 Sep 5 (pp. 14-25). Cham: Springer Nature Switzerland.

The authors should consider modifying their claims and also comparing with the method proposed in this paper.

2. Currently, the multi-target sequential adaptation section has very limited experiments. Given that the main claim of the paper is to reduce forgetting in the continual learning setting, the authors should show more experiments in the multi-target sequential adaptation setting to support the claims. The authors could show experiments with continual learning (1) on more domains and (2) on a different order of the same set of the domains.

**Strengths And Weaknesses:**

Strengths:

1. The problem statement is quite relevant considering that there’s always new data coming in and ensuring data privacy is important.

2. The authors re-implement previous approaches and compare with proposed approach in a unified framework.

3. The dual-speed optimization strategy and consistency learning helps mitigate forgetting.


Weaknesses:

1. The proposed method has limited technical contribution; consistency loss has been already used in the previous works and mutual information maximization is also a commonly used technique for regularization. The only novel part is the dual-speed update which helps with the issues of forgetting.

2. On various benchmarks, the proposed method rarely enhances both classification performance and forgetting issue (accuracy drop). In most cases it only performs better in one of the scenarios.

---

### Review · Reviewer_SLwC · 2025-03-06

**Summary Of Contributions:**

1. This paper reimplements previous  Source-Free Domain Adaptation (SFDA) approaches in a unified framework, evaluates them on four benchmarks, and identifies a trade-off between adaptation gain and forgetting loss.

2. This paper introduces a new problem setting in SFDA, namely Continual Source-Free Domain Adaptation (CoSDA), which aims to address the challenge of adapting to a sequence of target domains without accessing source data, while mitigating catastrophic forgetting of previously adapted domains.

3. This paper proposes a novel approach to address CoSDA with consistency regularization and a teacher-student dual-speed framework to balance adaptation and forgetting, incorporating Mixup and mutual information maximization for better generalization. This method performs comparably or slightly worse than baselines on several benchmarks but mitigates source performance drop more effectively.

**Audience:**

Yes

**Claims And Evidence:**

Yes

**Requested Changes:**

1. Introduce a clear metric to quantify the trade-off between adaptation gain and forgetting loss. Conduct additional experiments to analyze how this balance varies across different datasets to strengthen the evaluation of the method’s effectiveness.

2. Provide a detailed hyperparameter analysis, including justification for the chosen values through ablation studies or theoretical reasoning. Specifically, explain the selection of key hyperparameters such as $\tau$, $a$, $\alpha$, and $m$. Additionally, investigate the impact of batch size on the estimation of mutual information loss, as larger batch sizes generally improve distribution estimation accuracy. Experimental validation or theoretical insights on batch size selection should be included.

3. Clarify the notation for $ h_{\theta}(\mathbf{\tilde{X}}) $and $h_{\psi}(\mathbf{\tilde{X}})$, explicitly stating whether they represent logits or probabilities. If they correspond to logits, ensure that KL-divergence is computed between valid probability distributions to maintain mathematical consistency.

4. Align the method description with the implementation by explicitly stating that consistency loss is computed only for samples with confidence greater than $\zeta$, as reflected in `cosda.py` (line 10 and line 48). Additionally, clarify the effect of applying the confidence mask after Mixup, as it may not fully exclude low-confidence samples from influencing the final representations. Ensure that this design choice aligns with the intended purpose of the method.

Addressing points 2-4 is crucial for strengthening the paper and significantly influences my recommendation for acceptance.

**Strengths And Weaknesses:**

**Strengths**
1. The paper is well-structured, with a clear motivation. While there are some minor unclear descriptions in the method section, the overall presentation is easy to follow, and source code is provided for reproducibility.

2. This paper highlights an important but often overlooked aspect of domain adaptation—preserving source domain performance, rather than focusing solely on target adaptation. This is crucial since inference data can come from any domain.

3. This method proposed by this paper is easy to implement and compatible with existing SFDA approaches, allowing seamless integration to further improve performance and reduce forgetting.

4. This paper conducts well-designed experiments with multiple baselines and datasets, providing a comprehensive evaluation of the proposed method.

**Weaknesses**

1. This paper lacks an in-depth analysis of the adaptation-forgetting trade-off. While the paper discusses the trade-off between adaptation gain and forgetting loss, it does not establish a way to balance their importance. Consequently, when the experimental results show that the proposed method mitigates source domain performance drop but does not significantly improve target domain accuracy over baselines, it becomes difficult to assess the overall effectiveness of the model.

2. The method details lack clarity, particularly in the notation of  $h_{\theta}(\mathbf{\tilde{X}})$ and $h_{\psi}(\mathbf{\tilde{X}})$—it is unclear whether they represent logits or probabilities. If they correspond to logits, the use of KL-divergence raises concerns, as it is applied between a probability $\tilde{p} \in [0,1]$, and $h_{\psi}(\tilde{X}) $, which contains unbounded values ranging from negative to positive infinity. A clearer distinction is needed to ensure mathematical consistency.

3. There is an inconsistency between the source code and the method description. In `cosda.py` (line 10 and line 48), the implementation indicates that only data with confidence greater than $\zeta$ is used for computing the consistency loss. However, this detail is not explicitly mentioned in the method section or the algorithm table.  Additionally, there is a concern regarding the application of the confidence mask after Mixup. The mask is applied only to the original data, meaning that the mixed sample could still involve a low-confidence sample. This raises the question of whether the mask retains its intended effect in such cases.

4. The paper lacks necessary hype-rparameter analysis. It introduces three key hyper-parameters but does not provide experimental justification for their chosen values. Additionally, the hyper-parameters mentioned in Weakness 3 are not accompanied by explanations regarding their selection criteria. Furthermore, in the computation of the mutual information loss (Eq. 3), The batch size implicitly acts as a hyper-parameter because the paper estimates the data distribution using the empirical average over a mini-batch. Generally, a larger batch size leads to a more accurate distribution estimation. However, the paper does not provide experimental validation or theoretical analysis on how to choose an appropriate batch size.

---

### Note · Authors · 2025-03-14

**Comment:**

We sincerely thank all reviewers for their thoughtful and constructive feedback. After careful consideration, we recognize that our current manuscript contains significant limitations regarding experiments and analysis that would be difficult to address comprehensively within the available timeframe. Therefore, we respectfully request to withdraw this submission so that we may thoroughly revise our work to address these concerns and present a stronger contribution in the future.

**Withdrawal Confirmation:**

I have read and agree with the venue's withdrawal policy on behalf of myself and my co-authors.